# Similar excitability through different sodium channels and implications for the analgesic efficacy of selective drugs

Yu-Feng Xie[1], Jane Yang[1,2], Stéphanie Ratté[1], Steven A Prescott[1,2,3]*

[1]Neurosciences and Mental Health, The Hospital for Sick Children, Toronto, Canada; [2]Institute of Biomedical Engineering, University of Toronto, Toronto, Canada; [3]Department of Physiology, University of Toronto, Toronto, Canada

*For correspondence:
steve.prescott@sickkids.ca

**Competing interest:** The authors declare that no competing interests exist.

**Abstract** Nociceptive sensory neurons convey pain-related signals to the CNS using action potentials. Loss-of-function mutations in the voltage-gated sodium channel $Na_V1.7$ cause insensitivity to pain (presumably by reducing nociceptor excitability) but clinical trials seeking to treat pain by inhibiting $Na_V1.7$ pharmacologically have struggled. This may reflect the variable contribution of $Na_V1.7$ to nociceptor excitability. Contrary to claims that $Na_V1.7$ is necessary for nociceptors to initiate action potentials, we show that nociceptors can achieve similar excitability using different combinations of $Na_V1.3$, $Na_V1.7$, and $Na_V1.8$. Selectively blocking one of those $Na_V$ subtypes reduces nociceptor excitability only if the other subtypes are weakly expressed. For example, excitability relies on $Na_V1.8$ in acutely dissociated nociceptors but responsibility shifts to $Na_V1.7$ and $Na_V1.3$ by the fourth day in culture. A similar shift in $Na_V$ dependence occurs in vivo after inflammation, impacting ability of the $Na_V1.7$-selective inhibitor PF-05089771 to reduce pain in behavioral tests. Flexible use of different $Na_V$ subtypes exemplifies degeneracy – achieving similar function using different components – and compromises reliable modulation of nociceptor excitability by subtype-selective inhibitors. Identifying the dominant $Na_V$ subtype to predict drug efficacy is not trivial. Degeneracy at the cellular level must be considered when choosing drug targets at the molecular level.

## eLife assessment

This **fundamental** study provides an unprecedented understanding of the roles of different combinations of NaV channel isoforms in nociceptors' excitability, with relevance for the design of better strategies targeting NaV channels to treat pain. Although the experimental combination of electrophysiological, modeling, imaging, molecular biology, and behavioral data is **convincing** and supports the major claims of the work, some results remain inconclusive and need to be strengthened by further evidence. The work may be of broad interest to scientists working on pain, drug development, neuronal excitability, and ion channels.

## Introduction

Chronic pain affects between 11 and 40% of the population worldwide (*Cohen et al., 2021*). Neuropathic pain, which is pain arising from damage to the somatosensory nervous system, is particularly hard to treat with only 30% of patients achieving moderate (≥30%) relief using available treatments (*Finnerup et al., 2015*; *Rosenberger et al., 2020*). New treatments are needed but a meagre 11% of

analgesic drugs entering phase 1 trials are ultimately approved (*Hay et al., 2014*), triggering debate about why basic science discoveries are not yielding improved clinical outcomes (*Woolf, 2010*). Suggested explanations include flaws in preclinical animal testing (*Mogil, 2009*; *Taneja et al., 2012*) or clinical trial design (*Mao, 2012*) but biological explanations must also be considered. For example, degeneracy – the ability of a biological system to achieve similar or even equivalent function using different components (*Edelman and Gally, 2001*) – complicates modulation of neuronal excitability by allowing changes in diverse ion channels to potentially subvert the therapeutic effect of a drug targeting a particular channel (*Ratté and Prescott, 2016*). These explanations are not mutually exclusive but degeneracy continues to receive little consideration.

Like most neurons, nociceptive sensory neurons (nociceptors) rely on spikes to transmit information. Their excitability is thus critical for relaying information to the CNS. Nociceptor excitability is increased in many pathological pain conditions and the resultant increase in afferent input drives chronic pain (*Gold and Gebhart, 2010*; *Haroutounian et al., 2014*; *Yatziv and Devor, 2019*). Neuronal excitability depends on the complex interplay between diverse ion channels (*Alles and Smith, 2021*; *Bean, 2007*; *Waxman and Zamponi, 2014*) but some channels seem to be particularly important for pain. For instance, loss- or gain-of-function mutations in the gene *SCN9A*, which encodes the voltage-gated sodium channel $Na_V1.7$, cause congenital insensitivity to pain (CIP) or painful neuropathies, respectively (*Cox et al., 2006*; *Fertleman et al., 2006*; *Yang et al., 2004*; for review see *Dib-Hajj et al., 2013*). In rodents, nociceptor-specific deletion of $Na_V1.7$ abolishes acute and inflammatory pain (*Nassar et al., 2004*) but not neuropathic pain (*Nassar et al., 2005*; *Minett et al., 2014*). Neuropathic pain is blocked by deleting $Na_V1.7$ globally, including from sympathetic neurons (*Grubinska et al., 2019*; *Minett et al., 2012*), although not if the deletion is induced in adulthood (*Shields et al., 2018*). Furthermore, loss-of-function mutations in $Na_V1.7$ do not consistently reduce nociceptor excitability (see Discussion) and the associated insensitivity to pain involves increased opioid signaling (*MacDonald et al., 2021*; *Minett et al., 2015*), consistent with naloxone's ability to restore pain sensitivity in CIP patients (*MacDonald et al., 2021*; *Dehen et al., 1978*). These observations cast doubt on whether $Na_V1.7$ mutations produce CIP by reducing nociceptor excitability, pointing instead to a less direct mechanism that may be harder to reproduce pharmacologically.

Notwithstanding such reservations, several $Na_V1.7$-selective drugs have been developed (*Emery et al., 2016*; *Vetter et al., 2017*; *Yang et al., 2018*) but none have yet passed phase 2 clinical trials (*Kushnarev et al., 2020*; *Alsaloum et al., 2020*; *Eagles et al., 2022*; *Kitano and Shinozuka, 2022*). This has been attributed to poor target engagement (*Eagles et al., 2022*; *Mulcahy et al., 2019*; *Bankar et al., 2018*; *Kingwell, 2019*) yet prevention of the flare response by PF-05198007, a $Na_V1.7$-selective inhibitor, argues that at least some $Na_V1.7$ channels are blocked (*Alexandrou et al., 2016*). But CIP patients exhibit a normal flare response (*McDermott et al., 2019*), suggesting that their C fibers compensate for chronic loss of $Na_V1.7$ channels. Other $Na_V1.7$-selective inhibitors have struggled in phase 1 trials because of autonomic side effects (e.g. *Rothenberg et al., 2019*), as might be expected if those drugs block $Na_V1.7$ channels on sympathetic neurons, which is apparently necessary to prevent/reverse neuropathic pain (see above). But CIP patients exhibit normal autonomic function (*Cox et al., 2006*; *McDermott et al., 2019*), suggesting that their sympathetic neurons also compensate for chronic loss of $Na_V1.7$ channels. In those patients, might similar compensation occur in nociceptors and restore pain, only for that effect to be masked by enhanced opioid signaling (see above)? Descriptions of $Na_V1.7$ as 'the' threshold channel imply that it is irreplaceable for nociceptor excitability, consistent on the surface with pain insensitivity due to loss-of-function mutations in $Na_V1.7$ but inconsistent with some past electrophysiological data (*Flake et al., 2004*; *Zhang et al., 1997*). Clarifying whether nociceptors rely on $Na_V1.7$ is an unresolved issue important for predicting the analgesic efficacy of $Na_V1.7$-selective inhibitors.

A serendipitous observation prompted us to reassess the role of $Na_V1.7$ in nociceptor excitability and the implications for drug efficacy. Specifically, we observed that tetrodotoxin (TTX), which inhibits $Na_V1.7$ and several other TTX-sensitive (TTX-S) sodium channels, had variable effects in nociceptors, dramatically reducing their excitability in some conditions but not in others. This variability reveals that nociceptors can achieve similar excitability using different sodium channel subtypes, some of which are TTX-resistant (TTX-R). We demonstrate that a $Na_V1.7$-selective inhibitor produces analgesia only when nociceptor excitability relies on $Na_V1.7$. Insofar as increasingly selective drugs are more likely to

have their efficacy subverted by degeneracy, our results have profound yet underappreciated implications for target selection and drug development.

## Results

### Nearly equivalent excitability can arise from different voltage-gated sodium (Na$_V$) channel subtypes

Small dorsal root ganglion (DRG) neurons (soma diameter <25 µm) tend to spike repetitively when depolarized by current injection (**Amir et al., 1999**). In our sample, most small neurons genetically identified as nociceptors (see Materials and methods) spiked repetitively when tested 2–8 hr after dissociation (DIV0) or after 4–7 days in culture (DIV4-7), although the proportion of repetitively spiking neurons increased slightly over that interval ($\chi^2$=4.51, p=0.034, chi-square test; **Figure 1A**). Strikingly, 100 nM TTX had no effect on the spiking pattern at DIV0 but converted all but one neuron to transient spiking at DIV4-7. Amongst neurons that spiked repetitively at baseline, TTX reduced the firing rate and increased rheobase only at DIV4-7 (**Figure 1B**). TTX reduced spike height at DIV0 and DIV4-7, but more so at DIV4-7. There was a significant increase in capacitance and leak conductance density between DIV0 and DIV4-7, but no change in resting membrane potential (**Figure 1C**). Normalizing leak conductance by capacitance (which increases over time because of neurite growth) disambiguates whether changes in input resistance reflect changes in cell size or membrane leakiness. Consistent with current clamp data, voltage clamp recordings showed that only a small fraction of sodium current is TTX-S at DIV0, whereas nearly all sodium current was blocked by TTX at DIV4-7 (**Figure 1D**). Previous studies suggested that TTX-R channels play an important role in nociceptor excitability (**Caffrey et al., 1992**; **Renganathan et al., 2001**; **Rush et al., 2007**). Our initial results confirm this for DIV0 but show that their contribution diminishes after a few days in culture, with TTX-S channels becoming dominant by DIV4. Despite this reconfiguring of Na$_V$ channels, excitability was remarkably stable, consistent with previous work showing little change in excitability after axotomy despite large (but evidently counterbalanced) changes in TTX-R and TTX-S currents (**Flake et al., 2004**; **Zhang et al., 1999**). We show later that similar changes develop in vivo following inflammation with consequences for drug efficacy assessed behaviorally (see Figure 8), suggesting the Na$_V$ channel reconfiguration described above is not a trivial epiphenomenon of culturing.

### Different Na$_V$ channel subtypes control nociceptor excitability at DIV0 and DIV4-7

Next, we sought to identify the Na$_V$ subtype responsible for repetitive spiking at each time point, starting with DIV0. Of the TTX-R Na$_V$ channels expressed by nociceptors, Na$_V$1.8 has been implicated in repetitive spiking (**Renganathan et al., 2001**; **Rush et al., 2007**). We measured sodium current in voltage clamp before and after applying the Na$_V$1.8-selective inhibitor PF-01247324 (PF-24) (**Payne et al., 2015**). At DIV0, 1 µM PF-24 abolished most of the sodium current (**Figure 2A**). The PF-24-sensitive current had slow inactivation kinetics, like the TTX-R current and unlike the fast TTX-S current in **Figure 1D**, and consistent with previous descriptions of Na$_V$1.8 (**Vijayaragavan et al., 2001**). A different Na$_V$1.8 antagonist, A-803467 (**Jarvis et al., 2007**), had similar effects (**Figure 2—figure supplement 1**). In current clamp, PF-24 converted 7 of 8 repetitively spiking neurons to transient spiking and significantly reduced evoked spiking (**Figure 2B**). It also increased rheobase and decreased spike height but did not affect resting membrane potential (**Figure 2C**). PF-24 had negligible effects when tested at DIV4-7 (**Figure 2—figure supplement 2**). These results show that Na$_V$1.8 is the predominant Na$_V$ subtype at DIV0 and is necessary for repetitive spiking at that time point. To test the sufficiency of Na$_V$1.8 to produce repetitive spiking, we tuned a single-compartment, conductance-based model neuron (see Materials and methods) to reproduce DIV0 data described above. In this DIV0 model, inclusion of Na$_V$1.8 conductance was sufficient to generate repetitive spiking (**Figure 2D** left). The necessity of Na$_V$1.8 for repetitive spiking at DIV0 was also recapitulated: 85% reduction in the Na$_V$1.8 conductance converted spiking from repetitive to transient (**Figure 2D** and **Supplementary file 1A**).

Next, we sought to identify the Na$_V$ subtype responsible for repetitive spiking at DIV4-7 using PF-05089771 (PF-71) to inhibit Na$_V$1.7 (**Alexandrou et al., 2016**; **Theile et al., 2016**) and ICA-121431 (ICA) to inhibit Na$_V$1.1/1.3 (**McCormack et al., 2013**; **Strege et al., 2017**). Since Na$_V$1.1 is expressed mostly in medium-diameter (Aδ) neurons (**Osteen et al., 2016**) whereas Na$_V$1.3 is known to be

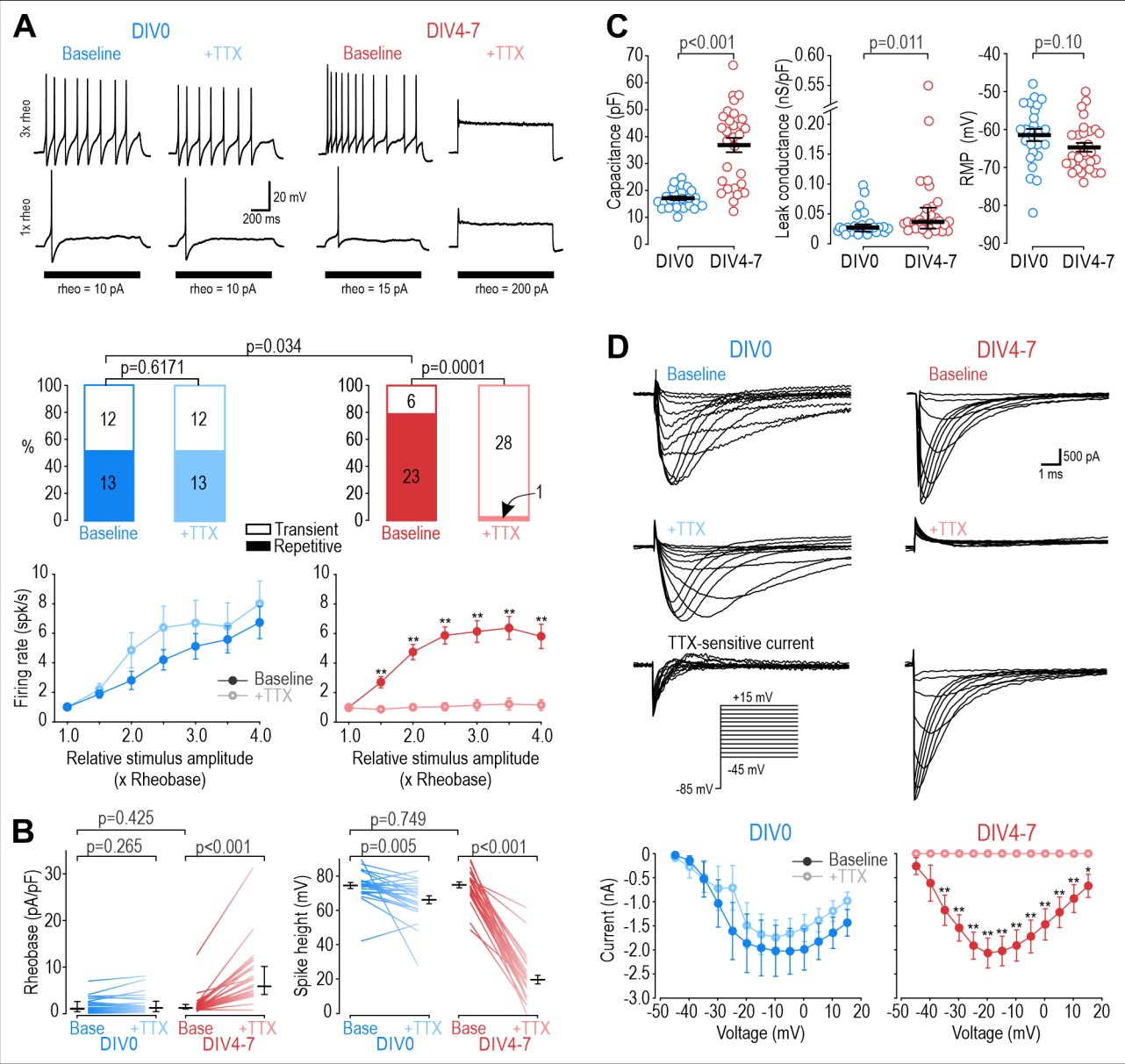

**Figure 1.** Different Na$_V$ subtypes produce similar excitability at different days in vitro (DIV). (**A**) Representative responses of small DRG neurons to current injection at rheobase and 3 x rheobase when tested on DIV0 (blue) or DIV4-7 (red) before (dark) and after (pale) bath application of 100 nM TTX. At DIV0, TTX did not alter spiking pattern ($\chi^2$=0.25, p=0.617, McNemar test) or significantly reduce firing rate (F$_{1,72}$=1.527, p=0.24, two-way repeated measure (RM) ANOVA; n=13). At DIV4-7, TTX significantly altered spiking pattern, converting all but one neuron to transient spiking ($\chi^2$=20.05, p<0.0001), and it significantly reduced firing rate (F$_{1,132}$=43.157, p<0.001, n=23). Only neurons with repetitive spiking at baseline are included in the firing rate plot. (**B**) At DIV0, TTX did not affect rheobase (Z$_{24}$=1.129, p=0.265, Wilcoxon rank test) but did reduce spike height (T$_{24}$=3.092, p=0.005, paired t-test). At DIV4-7, TTX increased rheobase (Z$_{28}$=4.681, p<0.001, Wilcoxon rank test) and dramatically reduced spike height (T$_{28}$=20.333, p<0.001, paired t-test). Notably, neurons at DIV0 and DIV4-7 did not differ in their baseline rheobase (U=316, p=0.425, Mann-Whitney test) or spike height (T$_{52}$=0.322, p=0.749, t-test). (**C**) Neurons at DIV0 and DIV4-7 differed in their total capacitance (T$_{52}$=6.728, p<0.001, t-test) and leak conductance density (U=216, p=0.011, Mann-Whitney test) but not in their resting membrane potential (T$_{52}$=1.668, p=0.101, t-test). (**D**) Sample voltage clamp recordings with command voltage stepped from –85 mV to +15 mV in 5 mV increments, before and after TTX. Sodium current was not significantly reduced by TTX at DIV0 (F$_{1,72}$=3.585, p=0.107, two-way RM ANOVA; n=7) but was completely abolished by TTX at DIV4-7 (F$_{1,108}$=33.526, p<0.001; n=10). Traces labeled 'TTX-sensitive current' represent the difference between current measured at baseline and after TTX, as determined by subtracting responses to the same voltage step under different pharmacological conditions. *, p<0.05; **, p<0.01; Student- Newman-Keuls post-hoc tests in A and D.

The online version of this article includes the following source data for figure 1:

**Source data 1.** Numerical values for data plotted in **Figure 1**.

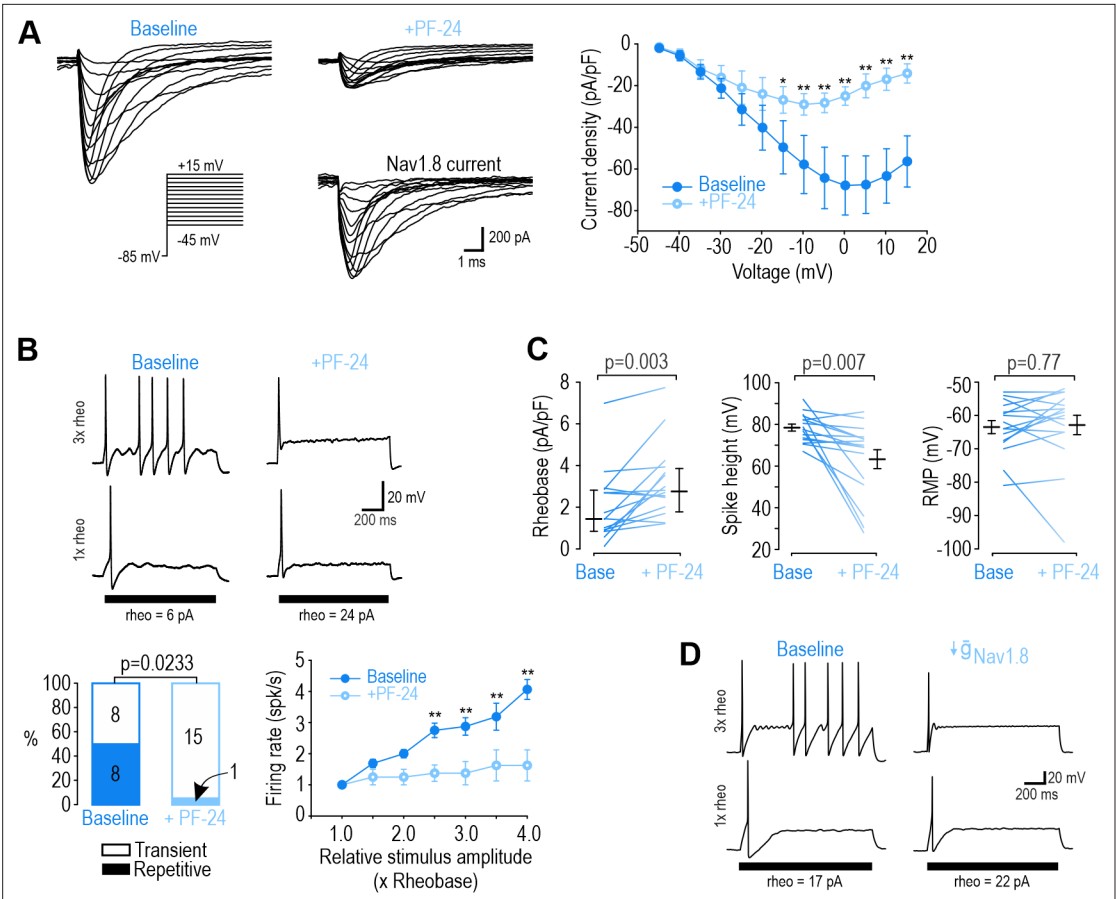

**Figure 2.** Na$_V$1.8 is necessary for repetitive spiking at DIV0. (**A**) Sample voltage clamp recordings show that sodium current was almost completely abolished by the Na$_V$1.8 inhibitor PF-24 (1 µM). Peak current was significantly reduced by PF-24 ($F_{1,72}$=12.651, p<0.012, two-way RM ANOVA; n=7). Traces labeled 'Na$_V$1.8 current' represent the difference between current measured at baseline and after PF-24, as determined by subtraction. Another Na$_V$1.8 inhibitor, A-803467, had a similar effect (see *Figure 2—figure supplement 1*). (**B**) PF-24 significantly altered spiking pattern ($\chi^2$=5.14, p=0.0233, McNemar test) and reduced firing rate ($F_{1,42}$=11.946, p=0.011, two-way RM ANOVA; n=8). (**C**) PF-24 significantly increased rheobase ($Z_{15}$=2.783, p=0.003, Wilcoxon rank test) and reduced spike height ($T_{15}$=3.151, p=0.007, paired t-test) but did not affect resting membrane potential ($T_{15}$=0.304, p=0.765, paired t-test). PF-24 had limited effects at DIV4-7 (*Figure 2—figure supplement 2*). (**D**) A computational model reproduced the effect of Na$_V$1.8 on spiking pattern (also see *Supplementary file 1A*). The PF-24 effect was simulated as a~85% reduction in Na$_V$1.8 ($\bar{g}_{Nav1.8}$ = 4 mS/cm2). *, p<0.05; **; p<0.01; Student-Newman-Keuls post-hoc tests in A and B.

The online version of this article includes the following source data and figure supplement(s) for figure 2:

**Source data 1.** Numerical values for data plotted in *Figure 2*, including supplements.

**Figure supplement 1.** Inhibiting Na$_V$1.8 at DIV0 with A-803467 had the same effect as PF-24.

**Figure supplement 2.** Inhibiting Na$_V$1.8 with PF-24 at DIV4-7 had negligible effects.

upregulated in C fibers after injury (for review, see *Bennett et al., 2019*), we ascribe the ICA effect to blockade of Na$_V$1.3. In voltage clamp, sodium current was significantly reduced by 30 nM PF-71, and most of the remaining current was blocked by 1 µM ICA (*Figure 3A*). In current clamp, each inhibitor (applied separately) converted a significant proportion of neurons to transient spiking and significantly reduced firing rate (*Figure 3B*). This argues that Na$_V$1.7 and Na$_V$1.3 are both necessary for repetitive spiking at DIV4-7. Inhibiting Na$_V$1.7 increased rheobase, unlike inhibiting Na$_V$1.3, and caused a stronger reduction in spike height (*Figure 3C*). Neither affected resting membrane potential. These results show that Na$_V$1.7 is the predominant Na$_V$ subtype at DIV4-7, but not the only one. PF-71 had negligible effects when tested at DIV0 (*Figure 3—figure supplement 1*). We re-tuned our computational model to reproduce DIV4-7 data, with both Na$_V$1.7 and Na$_V$1.3 being required to produce repetitive spiking, meaning neither channel is individually sufficient (*Figure 3D* and *Supplementary file 1A*). That said, inserting a higher density of either Na$_V$1.7 or Na$_V$1.3 could produce repetitive

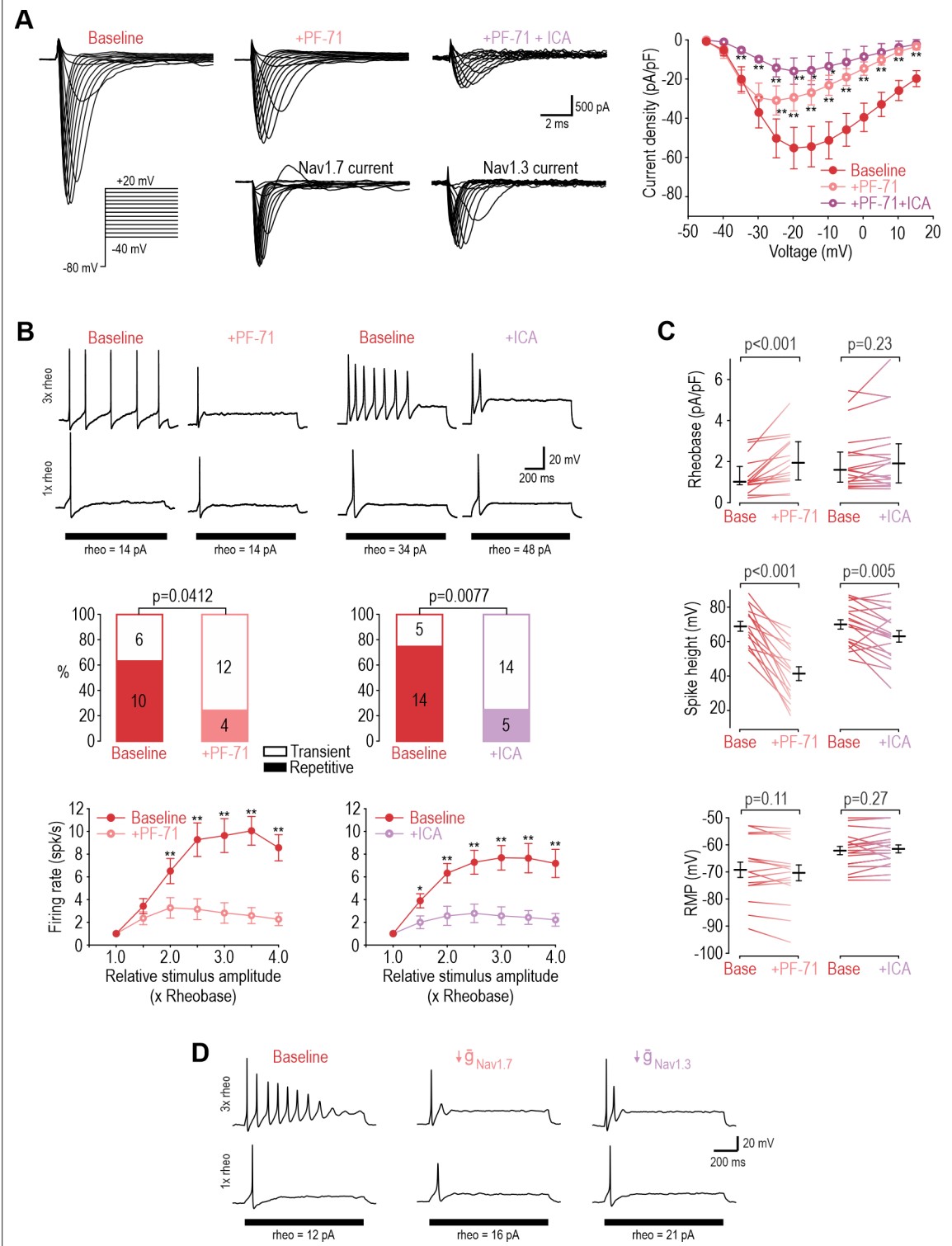

**Figure 3.** $Na_V1.3$ and $Na_V1.7$ are necessary for repetitive spiking at DIV4-7. (**A**) Sample voltage clamp recordings show that sodium current was reduced by the $Na_V1.7$ inhibitor PF-71 (30 nM) and by the $Na_V1.1/1.3$ inhibitor ICA (1 μM). Peak current was significantly reduced by PF-71 and ICA ($F_{2,192}=26.361$, $p<0.001$, two-way RM ANOVA; n=9). Traces labeled '$Na_V1.7$ current' and '$Na_V1.3$ current' represent the difference between current measured at baseline and after PF-72 and ICA, respectively, as determined by subtraction. (**B**) PF-71 and ICA both significantly altered spiking pattern ($\chi^2=4.17$, $p=0.041$ and $\chi^2=7.11$, $p=0.0077$, respectively, McNemar tests and significantly reduced firing rate) ($F_{1,54}=40.659$, $p<0.001$, n=10 and $F_{1,78}=35.156$, $p<0.001$, n=14, respectively, two-way RM ANOVAs). (**C**) PF-71 significantly increased rheobase ($Z_{18}=3.464$, $p<0.001$, Wilcoxon rank test) and decreased spike height ($T_{18}=7.946$, $p<0.001$, paired t-test). ICA did not significantly alter rheobase ($Z_{18}=1.248$, $p=0.225$) but did reduce spike height ($T_{18}=3.243$, $p=0.005$). Neither

*Figure 3 continued on next page*

*Figure 3 continued*

drug affected resting membrane potential ($T_{15}$=1.681, p=0.113 for PF-71; $T_{18}$=−1.132, p=0.272 for ICA, paired t-test). PF-71 had negligible effects at DIV0 (*Figure 3—figure supplement 1*). (**D**) A computational model reproduced the combined effects of Na$_V$1.3 and Na$_V$1.7 on spiking pattern (also see *Supplementary file 1A* and *Figure 3—figure supplement 2*). PF-71 effect was simulated as a 70% reduction in Na$_V$1.7 ($\bar{g}_{Nav1.7}$ = 10.5 mS/cm$^2$). ICA effect was simulated as a 90% reduction in Na$_V$1.3 ($\bar{g}_{Nav1.3}$ = 0.035 mS/cm$^2$). *, p<0.05; **, p<0.01; Student-Newman-Keuls post-hoc tests in A and B.

The online version of this article includes the following source data and figure supplement(s) for figure 3:

**Source data 1.** Numerical values for data plotted in *Figure 3*, including supplements.

**Figure supplement 1.** Inhibiting Na$_V$1.7 at DIV0 had negligible effects.

**Figure supplement 2.** Na$_V$1.7 and Na$_V$1.3 currents can compensate for each other.

spiking in the absence of the other subtype (*Figure 3—figure supplement 2*), consistent with Na$_V$1.7 and Na$_V$1.3 also being interchangeable.

## Acutely interchanging Na$_V$ subtypes does not affect spiking pattern

The ability of Na$_V$1.3, Na$_V$1.7 and Na$_V$1.8 to each encourage repetitive spiking is seemingly inconsistent with the common view that each Na$_V$ subtype contributes selectively to a different phase of the spike (for example, Figure 3 in *Bennett et al., 2019*). If Na$_V$1.8 were to activate exclusively at suprathreshold voltages, it could not initiate spikes and a different perithreshold-activating Na$_V$ channel would be needed, which is clearly inconsistent with our data. To verify that Na$_V$1.7 and Na$_V$1.8 currents are each sufficient to produce repetitive spiking, we tested whether the Na$_V$1.8 current necessary for spiking in our DIV0 computer model could be replaced with Na$_V$1.7, and whether the Na$_V$1.7 current necessary for spiking in our DIV4-7 computer model could be replaced with Na$_V$1.8. In both cases, repetitive spiking was restored after inserting the alternate current (*Figure 4A*). We

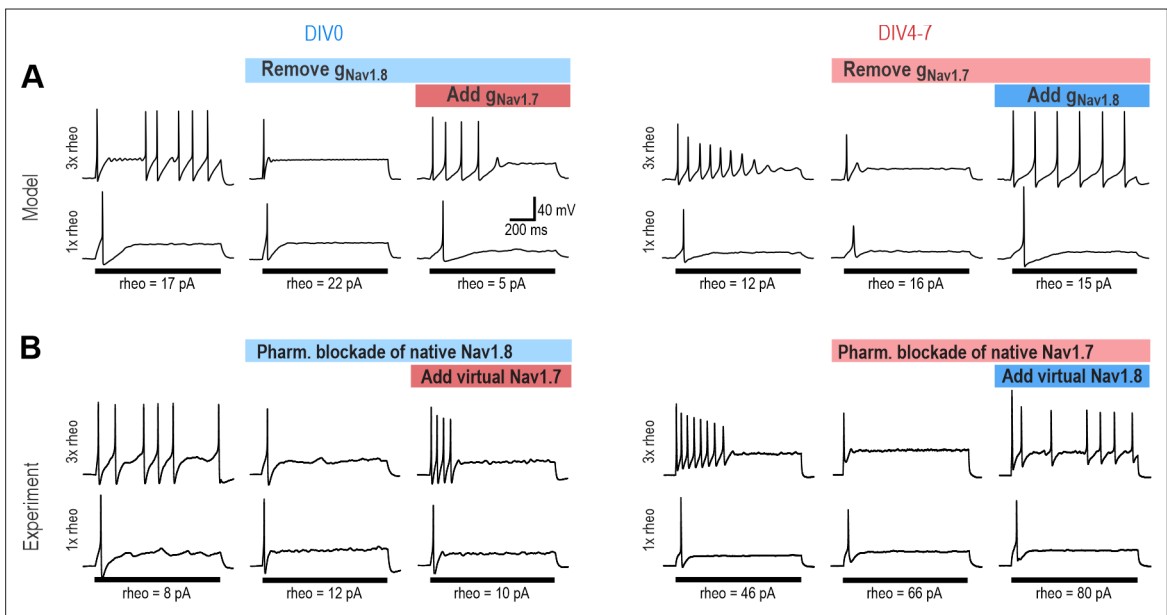

**Figure 4.** Na$_V$1.7 and Na$_V$1.8 are each sufficient to produce repetitive spiking in DIV0 and DIV4-7 neurons. (**A**) The computational model predicts that the Na$_V$1.8 conductance, which is "necessary" for repetitive spiking at DIV0 can, in principle, be replaced by Na$_V$1.7 (left), and vice versa at DIV4-7 (right). (**B**) Replacement experiments involved inhibiting native channels pharmacologically and then introducing virtual conductances using dynamic clamp. At DIV0 (left), inhibiting native Na$_V$1.8 (with PF-24) converted neurons to transient spiking, but introducing virtual Na$_V$1.7 reverted neurons to repetitive spiking (in 3 of 3 neurons tested). At DIV4-7, inhibiting native Na$_V$1.7 (with PF-71) converted the neuron to transient spiking, but introducing virtual Na$_V$1.8 reverted neurons to repetitive spiking (in 4 of 4 neurons tested). Repetitive spiking was likewise restored by replacing the blocked native channel with the corresponding virtual channel (*Figure 4—figure supplement 1*). Parameters for virtual channels were identical to simulations except for the maximal conductance density, which was titrated in each cell.

The online version of this article includes the following figure supplement(s) for figure 4:

**Figure supplement 1.** Virtual conductances restored repetitive spiking after pharmacological inhibition of the corresponding native conductance had converted the neuron to transient spiking.

then proceeded with equivalent experiments in real neurons, inhibiting Na$_V$1.8 with PF-24 on DIV0 or Na$_V$1.7 with PF-71 on DIV4-7, and then introducing the alternate channel virtually using dynamic clamp (see *Methods*). The replacement was successful in all neurons tested (*Figure 4B*). Inserting virtual Na$_V$1.8 after inhibiting native Na$_V$1.8 also restored repetitive spiking, and likewise for Na$_V$1.7 (*Figure 4—figure supplement 1*), verifying that our virtual channels were equivalent to the native channels we aimed to replace. Apart from maximal conductance density, which was titrated in each neuron, all other parameters used for dynamic clamp were identical to simulations. The success of dynamic clamp experiments helps validate our computational models insofar as virtual Na$_V$1.7 and Na$_V$1.8 currents interacted appropriately with native currents to produce repetitive spiking in real neurons, the same way they interact with other simulated currents in the model neuron. Please note that tests reported In *Figure 4B* involve replacing a native channel with a different virtual channel (e.g. native Na$_V$1.8 replaced with virtual Na$_V$1.7) whereas tests reported in *Figure 4—figure supplement 1* involve replacing a native channel with the equivalent virtual channel (e.g. native Na$_V$1.8 replaced with virtual Na$_V$1.8); the former demonstrates that Na$_V$ subtypes are interchangeable, whereas the latter serves as a positive control.

With the model neurons thus validated, we used simulations to infer Na$_V$1.7 and Na$_V$1.8 currents during different phases of the spike (*Figure 5A–D*). Since inward (depolarizing) current at voltages just below spike threshold is critical for spike initiation (*Prescott et al., 2008*), we sought to identify which Na$_V$ contributes to the subthreshold current. In the DIV0 model (*Figure 5A and B*), subthreshold inward current was mediated mostly by Na$_V$1.7 during the first spike (left) but by Na$_V$1.8 during the second and all subsequent spikes (right). We interpret this to mean that the first spike is initiated using Na$_V$1.7 whereas all subsequent spikes are initiated using Na$_V$1.8. This is explained by the small Na$_V$1.7 conductance at DIV0 quickly inactivating during the first spike and remaining inactive during subsequent spikes (*Figure 5—figure supplement 1A*). This is consistent with experimental results, where repetitive spiking at DIV0 was unaffected by inhibiting Na$_V$1.7 (see *Figure 1* and *Figure 3— figure supplement 1*) but was prevented by inhibiting Na$_V$1.8 (see *Figure 2*). Inactivation of Na$_V$1.7 after the first spike was reflected by an increase in voltage threshold between the first and second spike in the model (*Figure 5A*), which prompted us to check if the same increase was evident experimentally, which it was (*Figure 5E*). This unanticipated simulation result also predicted that TTX should affect the voltage threshold of the first spike in DIV0 neurons despite not having other notable effects (see *Figure 1*); as predicted, TTX caused a significant depolarizing shift in voltage threshold at DIV0 (*Figure 5—figure supplement 2*), further validating our model. In the DIV4-7 model (*Figure 5C and D*), subthreshold inward current was mediated by Na$_V$1.7 and Na$_V$1.3 during the first spike (left) and during all subsequent spikes (right), with Na$_V$1.8 contributing little. Even though inactivation reduced Na$_V$1.7 and Na$_V$1.3 current after the first spike (*Figure 5—figure supplement 1B*), those channels nonetheless provided sufficient inward current to support repetitive spiking at DIV4-7. Inactivation at DIV4-7 was reflected, however, in a combination of higher threshold and lower spike overshoot for the second spike, both in the model (*Figure 5C and D*) and in experiments (*Figure 5E*).

These results demonstrate that each Na$_V$ subtype does not contribute exclusively to a particular phase of the spike, and nor is each spike phase mediated exclusively by a particular Na$_V$ subtype. Instead, each Na$_V$ subtype contributes preferentially to a different spike phase depending on its voltage-dependency and on the conductance densities and inactivation status of other Na$_V$ subtypes; for instance, Na$_V$1.8 is often said to activate only at suprathreshold voltages, during the upswing of the spike, after the spike is initiated by Na$_V$1.7; but if Na$_V$1.7 is absent or inactivated, voltage threshold shifts into the Na$_V$1.8 activation range, thus enabling Na$_V$1.8 to activate at voltages that are now subthreshold. Indeed, a subtype's contribution can shift rapidly (because of channel inactivation) or slowly (because of changes in conductance density; see below).

## Changes in Na$_V$ subtype expression between DIV0 and DIV4-7

Next, we sought to identify the basis for the slow shift in which Na$_V$ subtype controls nociceptor excitability. *Figure 6A* shows mRNA levels for Na$_V$1.7 and Na$_V$1.8 relative to a housekeeping gene (left) and to each other (right). Na$_V$1.7 mRNA levels exceeded Na$_V$1.8 mRNA levels at both DIV0 and DIV7. Both decreased between DIV0 and DIV7, but Na$_V$1.8 more so, resulting in a significant decrease in the Na$_V$1.8:Na$_V$1.7 mRNA ratio. This pattern is consistent with the reduced role of Na$_V$1.8 at DIV4-7 but is inconsistent with the negligible role of Na$_V$1.7 at DIV0; specifically, we expected Na$_V$1.7 mRNA

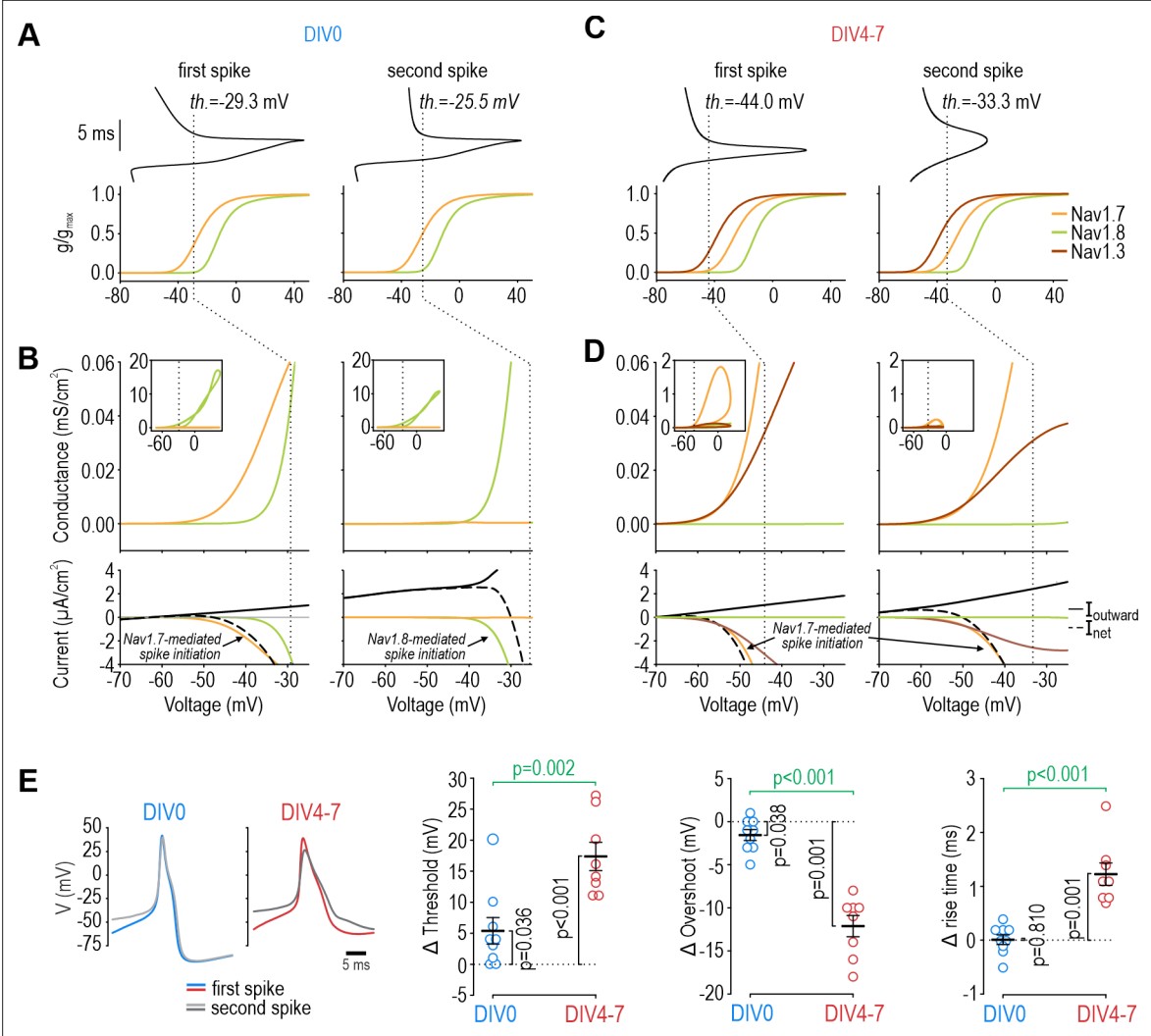

**Figure 5.** Contribution of Na$_V$1.7 and Na$_V$1.8 to spike initiation in DIV0 and DIV4-7 neurons. (**A**) Voltage (top) for first (left) and second (right) spikes in the DIV0 model aligned with voltage activation curves for each Na$_V$ subtype (bottom). Dashed line shows voltage threshold (defined as V where dV/ dt reaches 5 mV/ms). (**B**) Conductance plotted against voltage to create a phase portrait (top) showing Na$_V$ conductance at different phases of the spike. Inset shows full voltage range; main graph zooms in on voltages near threshold. Bottom plots show current plotted over the same voltage range. Whereas Na$_V$1.7 (orange) mediated nearly all perithreshold inward current for the first spike, voltage threshold increased – because Na$_V$1.7 inactivated (*Figure 5—figure supplement 1*) – and Na$_V$1.8 (green) mediated nearly all perithreshold inward current for the second spike. The unexpected contribution of Na$_V$1.7 to the first spike correctly predicted that TTX increases voltage threshold in DIV0 neurons (*Figure 5—figure supplement 2*). (**C, D**) In the DIV4-7 model, Na$_V$1.7 (orange) and Na$_V$1.3 (maroon) contributed to initiation of all spikes whereas the contribution of Na$_V$1.8 was negligible (due entirely to its low expression level). (**E**) Sample experimental traces showing differences in the first (blue/red) and second (grey) spikes at DIV0 and DIV4-7. Plots summarize differences (Δ) in threshold, overshoot potential, and spike rise time between 1st and 2nd spikes during repetitive spiking evoked by current injection. At DIV0, the 1st and 2nd spikes differ significantly in their threshold ($T_8$=2.522, p=0.036, one-sample t-test) and overshoot ($T_8$=0.038, p=0.038) but not rise time ($T_8$=0.249, p=0.810). At DIV4-7, the 1st and 2nd spikes differ in all measures (threshold: $T_7$=7.613, p<0.001; overshoot: $T_7$=−9.849, p<0.001; rise time: $T_7$=5.979, p<0.001). Statistical results (green) show that differences between 1st and 2nd spike at DIV4-7 are significantly larger than differences at DIV0 (threshold: $T_{15}$=−3.847, p=0.002; overshoot: $T_{15}$=7.922, p<0.001; rise time: $T_{15}$=−5.617, p<0.001, unpaired t-tests), consistent with our computational model.

The online version of this article includes the following source data and figure supplement(s) for figure 5:

**Source data 1.** Numerical values for data plotted in *Figure 5*, including supplements.

**Figure supplement 1.** Channel inactivation affects Na$_V$ subtype contribution on short timescale.

**Figure supplement 2.** Effect of TTX on voltage threshold in DIV0 neurons.

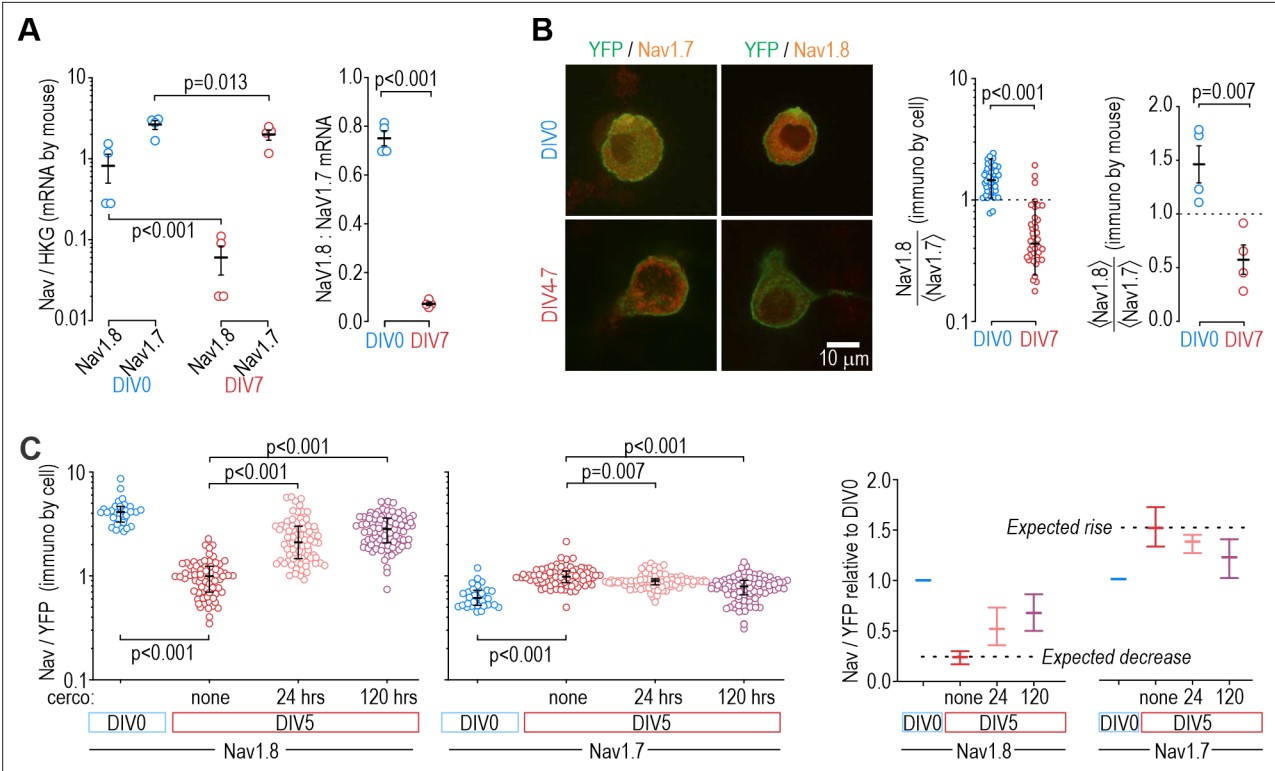

**Figure 6.** Protein levels, but not mRNA, reflect functional contributions of Na$_V$ subtypes at DIV0 and DIV7. (**A**) Both Na$_V$1.8 and Na$_V$1.7 mRNA levels (relative to a housekeeping gene (HKG), see Methods) decreased significantly between DIV0 and DIV4-7 (factor 1: time, $F_{1,12}$=56.677, p<0.001, factor 2: subtype, $F_{1,12}$=17.952, p=0.001, two-way ANOVA and Student-Newman-Keuls post-hoc tests on log transformed data, n=4 mice per time point) but more so for Na$_V$1.8 than for Na$_V$1.7 (interaction: time x subtype, $F_{1,12}$=11.455, p=0.005). The differential reduction yielded a significantly higher Na$_V$1.8: Na$_V$1.7 ratio at DIV0 than at DIV7 ($T_6$=21.375, p<0.001, unpaired t-test) but the increasing functional contribution of Na$_V$1.7 between DIV0 and DIV4-7 remains unaccounted for. (**B**) Immunoreactivity (IR) for Na$_V$1.8 protein exceeded Na$_V$1.7-IR at DIV0, but the opposite was true on DIV4-7, consistent with the functional contribution of each subtype. Na$_V$-IR was measured relative to YFP intensity in the same cell, and then each cell's Na$_V$1.8:YFP ratio was considered relative to the average Na$_V$1.7:YFP ratio in the co-processed coverslip (left) or average Na$_V$1.8:YFP ratio was considered relative to the average Na$_V$1.7:YFP ratio in the same animal (right). Ratios were >1 at DIV0 but decreased significantly at DIV4-7 (U=78, p<0.001, n=37 for DIV0, n=40 for DIV4-7, Mann-Whitney test (left) and $T_6$=4.046, p=0.007, unpaired t-test (right)). (**C**) Chronically applied cercosporamide (10 µM) mitigated changes in Na$_V$1.8- and Na$_V$1.7-IR at DIV5 (Na$_V$1.8: $H_3$=157.95, p<0.001; Na$_V$1.7: $H_3$=80.662, p<0.001; One-way ANOVA on ranks, Dunn's post-hoc tests, p<0.05 for all pairs). Data are summarized as median ± quartile. Panel on the right shows data normalized to baseline (DIV0) to emphasize relative changes. N=3 experiments.

The online version of this article includes the following source data for figure 6:

**Source data 1.** Numerical values for data plotted in *Figure 6*.

levels to increase between DIV0 and DIV7. Next, we investigated if functional changes were better reflected by changes in protein levels. Immunofluorescence for Na$_V$1.8 was higher than for Na$_V$1.7 at DIV0, and that ratio reversed at DIV7 (*Figure 6B*), consistent with functional changes. Moreover, cercosporamide (10 µM), a potent inhibitor of the eukaryotic translation Initiation Factor 4E (eIF4E), significantly mitigated the decrease in Na$_V$1.8 immunofluorescence and the increase in Na$_V$1.7 immunofluorescence when applied to cultured neurons for 24 or 120 hr prior to measurements on DIV5 (*Figure 6C*). Beyond showing that their mRNA levels do not correlate well with Na$_V$ contributions to nociceptor excitability, reminiscent of some previous work (e.g. *Berta et al., 2008*), these results suggest that translational regulation is crucial, although membrane trafficking and other downstream processes likely also contribute (*Dustrude et al., 2013*; *Yamane et al., 2017*). Specifically, our results suggest that there is not enough pre-existing Na$_V$1.7 channels that trafficking those channels to the membrane can explain observed functional changes; instead, synthesis of new Na$_V$1.7 protein from existing Na$_V$1.7 mRNA is involved, but that does not rule out changes in how Na$_V$1.7 protein is handled or why Nav1.8 decreases. Further investigation is required to explore those mechanisms.

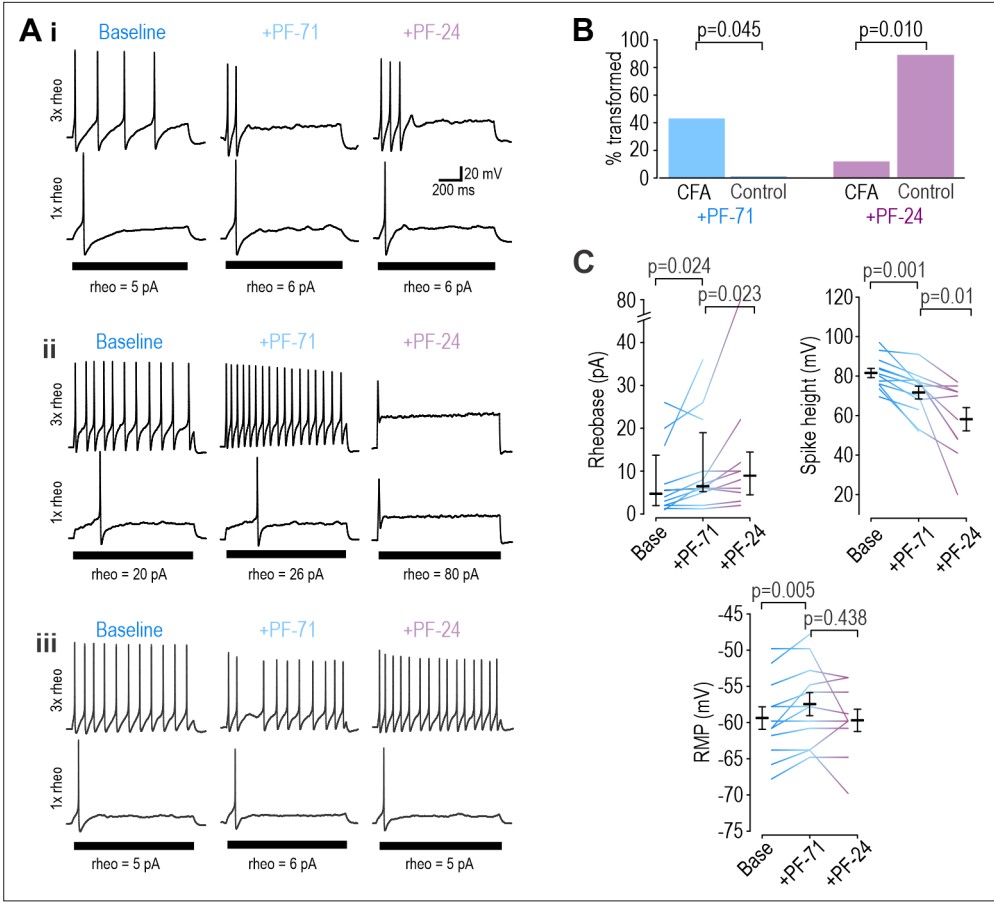

**Figure 7.** Inflammation alters Na$_V$ subtype contribution to nociceptor excitability. (**A**) Sample responses in DIV0 neurons from mice injected with CFA three days earlier. In 12 cells tested, PF-71 converted five neurons to transient spiking (**i**), encouraged repetitive spiking in four neurons (**ii**), and had no effect in three neurons (**iii**), thus highlighting increased heterogeneity after CFA. (**B**) At DIV0, the effect of PF-71 differed significantly between CFA and control neurons, converting 42% (5 of 12) CFA neurons from repetitive to transient spiking vs 0% (0 of 9) control neurons (p=0.045, Fisher Exact test). Applying PF-24 to neurons that continued to spike repetitively after PF-71 had little effect on CFA neurons, converting only 13% (1 of 7) of CFA neurons vs 88% (7 of 8) of control neurons (p=0.010, Fisher Exact test). Together these results argue that Na$_V$1.7 contributes more and Na$_V$1.8 contributes less to nociceptor excitability after inflammation. (**C**) At DIV0, PF-71 significantly increased resting membrane potential (T$_{11}$=−3.530, p=0.005, paired t-test) and rheobase (Z$_{11}$=2.186, p=0.024, Wilcoxon rank test), and significantly decreased spike height (T$_{11}$=4.413, p=0.001, paired t-test) in CFA neurons. Further addition of PF-24 significantly changed rheobase (Z$_9$=2.176, p=0.023, Wilcoxon rank test) and spike height (T$_9$=3.237, p=0.01, paired t-test) but did not affect resting membrane potential (T$_9$=1.049, p=0.321, paired t-test).

The online version of this article includes the following source data for figure 7:

**Source data 1.** Numerical values for data plotted in *Figure 7*.

## Analgesic efficacy of subtype-selective drugs depends on which Na$_V$ controls nociceptor excitability

If a Na$_V$1.7-selective inhibitor mediates analgesia by modulating nociceptor excitability, its analgesic efficacy hinges on nociceptor excitability being controlled by Na$_V$1.7. Accordingly, we predicted that the Na$_V$1.7-selective inhibitor PF-71 would have little if any effect on paw withdrawal under normal conditions, when Na$_V$1.8 controls nociceptor excitability (*Figure 2* and *Figure 3—figure supplement 1*), but would be effective if Na$_V$1.7 took over control. Inflammation increases Na$_V$1.7 channel trafficking and membrane expression (*Gould et al., 1998*; *Black et al., 2004*; *Liang et al., 2013*; *Akin et al., 2019*). To test if inflammation increased Na$_V$1.7's influence on nociceptor excitability, we recorded neurons acutely dissociated (DIV0) from DRGs of mice whose hind paw was injected

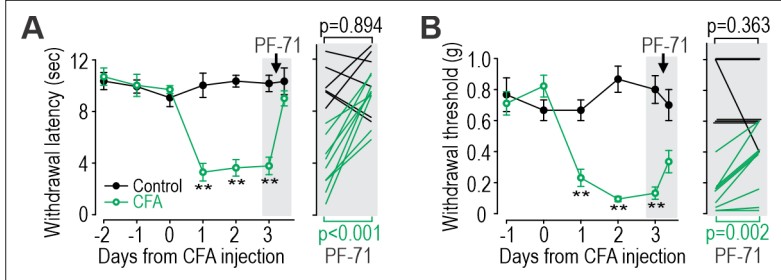

**Figure 8.** Inflammation-induced change in $Na_V$ subtype contribution impacts analgesic efficacy of PF-71. (**A**) CFA significantly increased thermal sensitivity ($F_{5,65}=19.556$, $p<0.001$, two-way RM ANOVA). PF-71 significantly decreased thermal sensitivity in mice injected 3 days prior with CFA ($T_8=-7.296$, $p<0.001$; paired t-test) but had no effect in naive mice ($T_5=-0.141$, $p=0.894$). (**B**) CFA significantly increased mechanical sensitivity ($F_{4,52}=16.786$, $p<0.001$). PF-71 significantly decreased tactile sensitivity in mice injected 3 days prior with CFA ($T_8=-4.341$, $p=0.002$) but had no effect in naive mice ($T_5=1.000$, $p=0.363$). Insets in both panels show values for each animal before and 2 hr after PF-71 injection. *, $p<0.05$; **, $p<0.01$; Student-Newman-Keuls post-hoc tests.

The online version of this article includes the following source data for figure 8:

**Source data 1.** Numerical values for data plotted in *Figure 8*.

with CFA 3 days prior. Sample traces in *Figure 7A* show that inflammation caused nociceptors to become much more variable in their reliance of specific $Na_V$ subtypes, presumably because neurons innervating the inflamed paw experience greater effects of inflammation than neurons innervating other, non-inflamed sites. Specifically, application of PF-71 to inhibit $Na_V1.7$ converted 5 of 12 (42%) CFA neurons to transient spiking vs 0 of 9 (0%) of control neurons, which is a significantly higher proportion ($p=0.0451$, Fisher exact test, *Figure 7B*, **left**), whereas subsequent application of PF-24 to inhibit $Na_V1.8$ converted only 1 of 7 (14%) of the remaining repetitive spiking CFA neurons to transient spiking vs 7 of 8 (88%) of control neurons, which is a significantly lower proportion ($p=0.010$, *Figure 7B*, **right**). PF-71 also significantly affected resting membrane potential, rheobase, and spike height after CFA (*Figure 7C*), unlike in control neurons (see *Figure 3—figure supplement 1*).

Results above confirm that $Na_V1.7$ takes on greater responsibility for nociceptor excitability after inflammation, which in turn predicts that PF-71 should reduce pain after inflammation but not under control conditions. As predicted, PF-71 significantly reduced thermal (*Figure 8A*) and tactile (*Figure 8B*) sensitivity in CFA-inflamed mice without having any effect in control mice. These results show that the inflammation-induced shift in $Na_V$ subtype expression, despite being variable (see *Figure 7*), is sufficient to cause a measurable change in drug efficacy assessed in vivo. Consistent with this, epigenetic repression of $Na_V1.7$ prevents/reverses hypersensitivity in inflamed and neuropathic mice without causing hyposensitivity in naïve mice (*Moreno et al., 2021*). This is unlike genetic deletion of $Na_V1.7$, which reduces thermal and tactile sensitivity in naïve mice (*MacDonald et al., 2021*), and with loss-of-function mutations in $Na_V1.7$ that abolish pain in humans (*Cox et al., 2006*). These inconsistencies rekindle concerns whether $Na_V1.7$ mutations, unlike pharmacological interventions, affect pain through mechanisms other than modulation of nociceptor excitability. Pharmacological reversal of hypersensitivity in chronic pain conditions (when $Na_V1.7$ is pathologically upregulated) without reducing normal nociceptive pain is clinically desirable, but this hinges on nociceptor hyperexcitability being $Na_V1.7$-dependent, which may be true of some but not all chronic pain conditions, or in only a subset of patients (*Baron and Dickenson, 2014*) (see Discussion).

## Discussion

Our results show that nociceptors can achieve similar excitability using different $Na_V$ channels. Whereas repetitive spiking depends on $Na_V1.8$ shortly after dissociation (*Figure 2*) and presumably under normal conditions in vivo, responsibility shifts to $Na_V1.7$ and $Na_V1.3$ after a few days in vitro (*Figure 3*). This is due to translationally regulated changes in $Na_V$ expression (*Figure 6*). Inflammation causes a similar shift in vivo (*Figure 7*). Importantly, acutely inhibiting a particular $Na_V$ is consequential (analgesic) only if that subtype is responsible for nociceptor excitability (*Figure 8*). This may explain why $Na_V1.7$-selective drugs have not performed well in clinical trials (see Introduction) – because

$Na_V1.7$ is not always necessary for nociceptor excitability depending on the expression level of $Na_V1.7$ and other $Na_V$ subtypes. Faster processes like channel inactivation also affect their relative contribution (*Figure 5*). These observations demonstrate the variable contribution of different $Na_V$ subtypes to nociceptor excitability. When unaccounted for, such variability can lead to inconsistencies at the root of poor reproducibility and translatability.

Although we have focussed here on $Na_V$ channels, numerous other channels are likely to change during culturing and in response to subtler, more clinically relevant in vivo insults like inflammation. Those changes occur through diverse mechanisms. Our results argue that transcriptional changes in $Na_V1.7$ do not account for its upregulation between DIV0 and DIV4-7, but transcriptional changes in other genes might nonetheless be important. And though our data implicate translational regulation, much more work is needed to work out the details. That work should consider the other ion channels and associated proteins (e.g. beta subunits) that interact with $Na_V$ channels, either physically or via mutual effects on membrane potential. In a degenerate system, one should ideally consider all the components contributing to the process of interest, but perfect is the enemy of good, which is to say that degeneracy should still be considered even with incomplete understanding of the system. Indeed, the interchangeability of $Na_V$ subtypes demonstrated here may help explain why subtype-specific drugs are not reliably effective against pain, even if myriad other still unidentified changes are also taking place.

Contrary to the view that certain ion channels are uniquely responsible for certain aspects of neuronal function, neurons use diverse ion channel combinations to achieve similar function (*Marder and Goaillard, 2006*; *O'Leary et al., 2014*). This degeneracy is crucial for enabling excitability and other aspects of neuron physiology to be homeostatically regulated by adjusting ion channels in response to perturbations (*Drion et al., 2015*; *Mishra and Narayanan, 2022*; *Yang et al., 2022*). Degeneracy also enables pathological changes in different ion channels to produce equivalent hyperexcitability (*Drion et al., 2011*). This is important insofar as similar excitability may belie differences in the underlying ion channels – differences that may render a neuron susceptible or impervious to a drug depending on the functional necessity of the targeted ion channel in that neuron. This is precisely what our data demonstrate in nociceptors. Similar observations have been made in substantia nigra neurons, whose pacemaker activity can be mediated by $Na_V$ channels or by voltage-gated calcium channels, meaning TTX may or may not block their spiking (*Drion et al., 2011*; *Puopolo et al., 2007*). Similar interchangeability is evident for the burst firing of Purkinje neurons (*Swensen and Bean, 2005*).

Degeneracy also exists at the circuit level (*Knox et al., 2018*; *Prinz et al., 2004*), where it allows differences in the intrinsic excitability of component neurons to be offset (and effectively hidden) by differences in synaptic weights (*Grashow et al., 2010*). Relevant for pain processing, the spinal dorsal horn circuit can achieve similar output using different synaptic weight combinations (*Medlock et al., 2022*); specific neuron types may have a greater or lesser impact on circuit function depending on those weights. In effect, degeneracy introduces contingencies. The role of any ion channel in a neuron (or any neuron in a circuit) depends on the other ion channels in that neuron (or the synaptic connections with other neurons in the circuit). Because of such contingencies, a drug may engage its target without producing the intended cellular, circuit or clinical effect. Indeed, different combinations of $GABA_A$ receptor activation and chloride driving force can produce equivalent synaptic inhibition (*Prescott et al., 2006*), but when inhibition is incompensably compromised, the underlying cause necessitates different interventions (*Lee and Prescott, 2015*). By this logic, if upregulation of $Na_V1.7$ is responsible for nociceptor hyperexcitability after nerve injury or inflammation, $Na_V1.7$ is an ideal target since 'normal' neurons (not reliant on $Na_V1.7$) would be spared the effects of a $Na_V1.7$-selective drug, but the long-term efficacy of such a drug hinges on hyperexcitability remaining $Na_V1.7$-dependent, which cannot be assumed (*Ratté and Prescott, 2016*). Furthermore, if myelinated afferents (which express minimal $Na_V1.7$ *Djouhri et al., 2003*) are responsible for mechanical allodynia under neuropathic conditions (*Campbell et al., 1988*; *Koltzenburg et al., 1992*; *Liu et al., 2000b*; *Liu et al., 2000a*), then $Na_V1.7$-selective drugs should not be expected to alleviate that symptom, which evidently they do not (*Shields et al., 2018*), at least not through a direct mechanism. Indeed, ablating nociceptors abolishes acute and inflammatory pain but not neuropathic pain (*Minett et al., 2014*; *Abrahamsen et al., 2008*). Pathological pain being mediated by more than one afferent type is another example of circuit-level degeneracy.

To be interchangeable, Na$_V$ subtypes must functionally overlap (*Goaillard and Marder, 2021*; *Yang and Prescott, 2023*). Indeed, Na$_V$1.8 and Na$_V$1.7 are similar but not identical in their gating properties; for example, their voltage-dependencies partially overlap but the activation curve for Na$_V$1.8 is right-shifted compared to Na$_V$1.7 (*Schild and Kunze, 1997*). Consequently, Na$_V$1.7 activates at voltages near threshold whereas Na$_V$1.8 tends to activate at suprathreshold voltages, during initiation and upstroke of the spike, respectively (*Alsaloum et al., 2020*; *Bennett et al., 2019*). But that separation is not absolute. We found that Na$_V$1.7 contributes to initiation of the first spike in DIV0 neurons, but because it inactivates more readily than Na$_V$1.8, initiation of all subsequent spikes depends on Na$_V$1.8 (see *Figure 5*), which activates at perithreshold voltages because voltage threshold is high (depolarized) in the absence of Na$_V$1.7. At DIV4-7, Na$_V$1.7 still inactivates (which causes voltage threshold to rise) but, because of its higher density, continues to produce enough inward current to continue to initiate later spikes. The activation pattern we report for the first spike at DIV0 is consistent with *Blair and Bean, 2002* who quantified the contribution of different Na$_V$ channels by recording pharmacologically isolated currents while varying the holding potential according to the spike waveform. Our results go further in showing how responsibilities shift across different spikes within a train (because of differential Na$_V$ inactivation) and across conditions (because of changes in Na$_V$ expression). Although we focused on Na$_V$ channels in this study, other ion channels are likely also undergoing changes; indeed, changes in the AHP shape between DIV0 and DIV 4–7 (see *Figure 1A*) point to changes in potassium channels. All neurons presumably regulate their excitability in a degenerate manner but likely do so by adjusting different sets of ion channels. Even within a given neuron type, the identify of adjusted ion channels may depend on the nature of the perturbation or on a multitude of other factors.

With respect to reproducibility, labs testing nociceptors after different times in vitro would be expected to reach contradictory conclusions about the relative importance of a given Na$_V$ subtype. Likewise, a testing protocol focusing on single spikes (the equivalent to the first spike in a train) would yield different results from one that considers repetitive spiking. Along the same lines, voltage clamp protocols that deliberately hold membrane potential at unnaturally hyperpolarized voltages to relieve inactivation before stepping up the voltage can give a misleading impression of how much a Na$_V$ subtypes contributes under natural conditions (i.e. with natural levels of inactivation). Such discrepancies might be chalked up to irreproducibility if the consequences of those methodological differences are not appreciated, especially if one overlooks how degeneracy allows responsibilities to shift between ion channels. Indeed, the pain literature is replete with apparent inconsistencies. We would argue that most of those studies are correct, but only under limited conditions. Failure to identify and report those conditions (contingencies) represents a huge impediment to translation. A recent review of degeneracy in epilepsy (*Stöber et al., 2023*) reveals many similarities with chronic pain. Observations that effective antiseizure medications often act on multiple targets and that patients with the same type of epilepsy respond heterogeneosly to a given treatment offer circumstantial evidence of degeneracy, but more deliberate testing is required to quantify degeneracy and its impact.

In summary, our results show that nociceptors can achieve similar excitability using different Na$_V$ subtypes. The importance of a given subtype can shift on long and short timescales, yielding results that are seemingly inconsistent. By elucidating those shifting responsibilities, our results highlight the degenerate nature of nociceptor excitability and its functional implications. Degeneracy makes it impossible to claim without reservation that a particular Na$_V$ subtype is uniquely responsible for pathological pain. Greater appreciation of degeneracy's implications would prompt better experimental design, more cautious interpretation, and, ultimately, improved translation.

# Materials and methods

## Animals

All animal procedures were approved by the Animal Care Committee at The Hospital for Sick Children (protocol #53451) and were conducted in accordance with guidelines from the Canadian Council on Animal Care. We used the Cre-*lox*P recombinase system to generate mice that express ChR2-eYFP in Na$_V$1.8-expressing neurons. Mice were obtained by crossing homozygous Ai32 mice (B6.Cg-Gt(ROSA)26Sortm32(CAG-COP4*H134R/EYFP)Hze/J) from Jax (#012569), which express ChR2(H134R)-eYFP in the presence of Cre recombinase, with Na$_V$1.8-Cre mice (Tg(Scn10a-cre)1Rkun),

which express Cre recombinase in $Na_v1.8$-expressing neurons (kindly provided by Rohini Kuner). These neurons are primarily nociceptive and thermoreceptive (*Agarwal et al., 2004*). The $Na_v1.8$ promoter leads to transgene expression in >90% of neurons expressing markers of nociceptors (*Nassar et al., 2004*). To ensure that our transgenic mice were typical of wild-type mice with the same background (C57BL/6 j), experiments reported in *Figure 1* were repeated in both genotypes for comparison. There was no effect of genotype on rheobase, spike height, input resistance, or spiking pattern, nor was there any significant interaction between genotype and effects of TTX except for spike height at DIV4-7, where TTX had a marginally larger effect in wild-type mice (Two-way ANOVA, $F_{1,54}=4.968$, p=0.03, see source data file); therefore, we pooled the data for *Figure 1*. Having verified that our foundational observations held across different genotypes, we used transgenic mice for all subsequent experiments in order to identify eYFP-expressing nociceptors for patching, collection, or imaging.

## Dorsal root ganglia neuron cultures

All key reagents are listed in *Supplementary file 1B*. Methods for primary DRG culture have been described previously (*Malin et al., 2007*). Briefly, adult mice (>7-week-old) were anaesthetised with isoflurane and perfused intracardiacally with cold HBBS (without Ca and Mg, LifeTech 14170112) supplemented with (in mM) 15 HEPES, 28 Glucose, 111 sucrose, and pH adjusted with NaOH to 7.3–7.4; osmolarity 319–321. Lumbar dorsal root ganglia (DRGs) were extracted (L2-5, except for CFA-inflamed mice, in which we only took L4), digested with papain (Worthington Biochemical Corp.) and collagenase (Worthington Biochemical Corp.)/dispase II (Sigma), and mechanically dissociated by trituration before being plated onto poly-D lysine-coated coverslips and incubated in Neurobasal media (Gibco 21103–049) supplemented with 1% fetal bovine serum (FBS), B-27 supplement (Thermo Fisher 17504–044) and 0.5 mM L-Glutamine (Gibco 25030–081) for an initial period of 2 hr. After this, media was changed to maintenance media (same as plating media but without FBS) and cells were maintained in a 5% $CO_2$ incubator at 37 °C. Media was changed every 3–4 days thereafter. Neurons were recorded at two time points after plating: 2–8 hr (referred to as DIV0) or 4–7 days (referred to as DIV4-7). Neurons were tested at intermediate time points (DIV1-3) in the exploratory phase of our study but cellular heterogeneity prevented a clear picture of their TTX-sensitivity, presumably because different neurons shift from $Na_v1.8$ to $Na_v1.7$ at different rates. By DIV4, TTX sensitivity had stabilized, and so recordings from DIV4-7 were pooled for comparison with recordings on DIV0. When tests were conducted on a specific day within the DIV4-7 range, the specific DIV is reported but should be considered representative of the DIV4-7 range. Additional experiments are required to determine the exact time course of the $Na_v$ switch and its mechanistic basis; to mitigate effects of cellular heterogeneity, this would ideally involve techniques that allow longitudinal measurements in the same cell.

## Electrophysiology

Coverslips with cultured neurons were transferred from the incubator to a recording chamber perfused with artificial cerebrospinal fluid containing (in mM): 126 NaCl, 2.5 KCl, 2.0 $CaCl_2$, 1.25 $NaH_2PO_4$, 26 $NaHCO_3$, 2 $MgCl_2$, and 10 glucose, bubbled with carbogen (5% $CO_2$:95% $O_2$) at room temperature. Neurons were visualized with gradient contrast optics on a Zeiss AxioExaminer microscope using a 40 x, 0.75 NA water immersion objective (N-Achroplan, Zeiss) and IR-1000 Infrared CCD camera (Dage-MTI). YFP expression was visualized by epifluorescence (X-Cite, Excelitas) using a Zeiss filter set (46HE). A long-pass filter (OG590) was positioned in the transmitted light path to avoid activating ChR2 while patching. No optogenetic testing was performed as part of this study. Cells expressing YFP and with a soma diameter <25 μm were targeted for whole cell recording using pipettes (~5 MΩ resistance) pulled from borosilicate glass (WPI). Neurons were tested from at least 3 different mice for each condition.

For current clamp recordings, pipettes were filled with intracellular solution containing (in mM): 140 K-gluconate, 2 $MgCl_2$, 10 HEPES, 0.2 EGTA, 3.8 Na-ATP and 0.4 Na-GTP with pH adjusted to 7.3 with KOH; osmolarity was ~300 mOsm. A liquid junction potential correction of 15 mV was applied to all reported voltages. Series resistance was compensated to >70%. Signals were amplified with an Axopatch 200B amplifier (Molecular Devices, Sunnyvale, USA), low-pass filtered at 2 kHz, digitized with a Power1401 A/D device (Cambridge Electric Design, Cambridge, UK), and recorded at 10 kHz using CED software Signal version 6. After the natural resting membrane potential was noted, neurons were adjusted to –70 mV using continuous current injection in current clamp mode. Action potentials

(spikes) were evoked using a series of 1 s long depolarizing current injections. Rheobase was defined as the minimal current required to evoke a spike. Neurons were tested with current injections from 1 x rheobase to 4 x rheobase using increments of 0.5 x rheobase. Repetitive spiking neurons were defined as those producing ≥3 spikes in response to any stimulus intensity; transient spiking neurons consistently produced ≤2 spikes. Spike threshold was defined as voltage where dV/dt first exceeds 5 mV/ms (*Davidson et al., 2014*). Spike height was measured from threshold to peak of the action potential. Only neurons with a resting membrane potential below –50 mV, spikes overshooting 0 mV and recordings with <20% change in series resistance were tested and analyzed. For dynamic clamp experiments, the pipette shank was painted with Sylgard (Dow) to reduce pipette capacitance. Virtual $Na_V1.7$ and $Na_V1.8$ conductances were introduced into the cells using CED software Signal v6. Currents were defined using the Hodgkin-Huxley equation, using the same parameter values as in our computational model (see below).

For voltage-clamp recordings, the bath solution was adjusted to reduce sodium currents to ensure proper clamping (*Liu et al., 2000a*). Bath solution contained (in mM): 65 NaCl, 50 choline chloride, 5 KCl, 5 HEPES, 5 $MgCl_2$, 10 glucose, and 0.1 $CaCl_2$, plus 0.1 $CdCl_2$ to block calcium currents, and 20 TEA and 5 4-AP to block potassium currents; pH was adjusted to 7.4 with NaOH. Pipettes were filled with intracellular solution containing (in Mm): 140 CsCl, 10 HEPES, 2 $MgCl_2$, 1 EGTA, 3.8 Na-ATP, 0.4 Na-GTP; pH was adjusted to 7.3 with CsOH. The resulting pipette resistance was ~3 MΩ. A liquid junction potential correction of 4.8 mV was applied to all command voltages. Sodium currents were recorded during 20 ms-long steps from –85 mV to voltages between –45 and +15 mV. Series resistance was compensated to >80%. Signals were amplified, low-pass filtered at 5 kHz, and digitized as described for current clamp recordings.

For all in vitro pharmacology, drugs were bath applied at a concentration chosen to selectively block the $Na_V$ subtype of interest based on published EC50 values (*Supplementary file 1C*).

## Quantitative reverse transcription PCR (RT-qPCR)

Cultured DRG neurons <25 µm and expressing eYFP were identified as described above for patching. Coverslips were perfused with aCSF made with DEPC-treated $ddH_2O$, and identified neurons were collected using a glass pipette filled with intracellular solution also made from DEPC-treated $ddH_2O$ (composition otherwise the same as described above for electrophysiology). Approximately 50 neurons were collected at DIV0 and at DIV4-7. Total mRNA was extracted with a PureLink RNA mini kit after digestion of genomic DNA with DNase I (Thermo Fisher Scientific) and the cDNA was synthesized with a SuperScript II first-strand synthesis kit (Thermo Fisher Scientific) according to instructions. RT-qPCR was performed with the cDNA primers of target genes (*Supplementary file 1D*), and the PowerUp SYBR Green master mix (Thermo Fisher Scientific) in the QuantStudio-3 real-time PCR system. The primers were designed with IDT and spanned at least one exon longer than 1000 bp in order to exclude contamination from genomic DNA. Non-RT mRNA was also used as a negative control to exclude contamination from genomic DNA. All target genes were performed in triplicate for each sample and the experiments were repeated in at least 3 separate batches of cells (i.e. three biological replicates). $Na_V1.7$ and $Na_V1.8$ transcript levels were analyzed using the 2-ΔΔCT method and compared with the housekeeping gene HPRT.

## Immunocytochemistry

Cultured DRG neurons were treated with 4% paraformaldehyde for 10 min, rinsed 3 x with cold PBS, and permeabilized with 0.1% Triton X-100 in PBS. After another 3 x rinse with PBS, neurons were treated with 10% normal goat serum for 30 min followed with rabbit primary $Na_V1.7$ antibody (1:200, Alomone, ASC-008, RRID:AB_2040198) or $Na_V1.8$ antibody (1:200, Alomone, ASC-028, RRID:AB_2341070) in PBS with 0.1% Tritween-20 and 1% BSA for 1 hr. For some of the coverslips, primary antibodies were replaced with control peptides (ASC008AG1040 for $Na_V1.7$ and ASC016AG0640 for $Na_V1.8$) provided by Alomone as negative controls. Following 3 x rinse in PBS, neurons were incubated in the dark with goat anti-rabbit secondary antibody Alexa Fluor-647 (1:500, Abcam) in PBS containing 1% BSA for 1 hr, followed by DAPI staining for 10 min. All incubations were done at room temperature. Finally, coverslips were mounted on slides with mounting media (Abcam, ab128982), imaged with a spinning disk confocal microscope (Quorum Technologies) using the same acquisition setting across all imaging sessions, and analyzed with Volocity software (v6.5.1). Protein levels are measured using fluorescence

intensity and expressed relative to each other (e.g. ratios in *Figure 6C*) or relative to fluorescence intensity for YFP in the same cells. Each condition was tested in a minimum of three animals (i.e. three biological replicates).

## Behavioral testing

Behavioral tests were performed on adult mice (male and female, 8–12 weeks). Mice were acclimated to the testing environment for at least 1 hr the day prior to start of experiments. Behavioral testing (von Frey test and Hargreaves test) was then performed for 2–3 consecutive days for baseline and for another 3 days after CFA injection. Behavioral tests were performed at the same time in the morning, at room temperature (21 °C) following a 1-hr acclimation period. Animals were randomly assigned to experimental groups and the experimenter was blind to the drug condition. None of the tested animals was excluded from analysis.

### CFA injection

CFA (Sigma, F5881) was thoroughly dissolved in saline (1:1) by vortexing the mixture. The resulting CFA solution (20 µl) was injected subdermally into the left hind paw under light isoflurane anaesthesia. The injection was performed shortly after the last baseline test, on Day 0.

### PF-71 administration

Injectable PF-71 solution was prepared by first dissolving PF-71 in DMSO to make a 5% stock solution; dissolution was achieved by heating to 37 °C and vortexing. On the day of injection, stock solution was dissolved in sunflower oil (5% v/v) by sonicating for 5 min. Freshly prepared final PF-71 solution was injected intraperitoneally (1 g/kg body weight). Behavioral testing was performed 2 hr after injection of PF-71 or vehicle.

### Von Frey testing

Mechanical hyperalgesia was assessed with von Frey filaments (North Coast) using the SUDO method (*Bonin et al., 2014*). The average of three trials in each animal was used for analysis.

### Hargreaves testing

Thermal hyperalgesia was assessed with the Hargreaves apparatus (Ugo Basile, Italy). Radiant heat was applied to the plantar surface of the left hind paw. Interval between stimulus onset and paw withdrawal was defined as paw withdrawal latency (PWL). A 20 s cut-off prevented damage to the skin if the animal failed to withdraw. The average of 3 trials in each animal was used for analysis.

## Statistical analysis

Results are expressed as mean ± SEM when data are normally distributed or otherwise as median and quartiles. Normality was tested using the Kolmogorov-Smirnov test. Analysis was performed with GraphPad Prism (v9) and SigmaPlot (v11). Normally distributed data were compared using t-tests or two-way ANOVAs followed by a Student Newman-Keuls post hoc test. Non-normally distributed data were compared using Mann-Whitney and Wilcoxon signed rank tests. Fisher exact and McNemar test were used for categorical data. Exact significance values and test results are reported throughout figure legends.

## Computer model

Two separate, single compartmental models were built for DIV 0 and DIV4-7. They have the same, seven conductances: $g_{\text{Nav1.3}}$, $g_{\text{Nav1.7}}$, $g_{\text{Nav1.8}}$, $g_{\text{Kdr}}$, $g_{\text{M}}$, $g_{\text{AHP}}$ and $g_{\text{Leak}}$. Channel equations and their gating parameters are provided in *Supplementary file 1E*. Conductance densities at baseline (*Supplementary file 1F*) were tuned to qualitatively reproduce the changes in $\text{Na}_{\text{V}}$ channel expressions at DIV 0 and 4–7 indicated by the experiments; changes to other channels were minimized between the two models. The effects of ICA, PF-71 and PF-24 were simulated by adjusting $\bar{g}_{\text{Nav1.3}}$, $\bar{g}_{\text{Nav1.7}}$ and $\bar{g}_{\text{Nav1.8}}$, respectively, as reported in the figures. To re-introduce voltage noise that is otherwise missing from simulations, we included an Ornstein-Uhlenbeck process with $\mu_{\text{noise}}$ = 0 $\mu\text{A/cm}^2$, $\sigma_{\text{noise}}$ = 0.05 $\mu\text{A/cm}^2$, and $\tau$ = 5ms. All simulations were conducted in MATLAB using the forward Euler integration

method and a time step of 0.05–0.1ms. Computer code is available at ModelDB (https://modeldb.science/2016663) and in *Source code 1*.

## Acknowledgements

This work was supported by a Restracomp fellowship to JY and by a Canadian Institutes of Health Research Foundation Grant (FDN167276) to SAP. We thank Rohini Kuner for providing Na$_v$1.8-Cre mice, Jason Jeong and Russell Smith for expert technical assistance with animal care and cell cultures, and Yongqian Wang for advice on qPCR data acquisition.

## Additional information

### Funding

| Funder | Grant reference number | Author |
| --- | --- | --- |
| Canadian Institutes of Health Research | FDN167276 | Steven A Prescott |

The funders had no role in study design, data collection and interpretation, or the decision to submit the work for publication.

### Author contributions

Yu-Feng Xie, Conceptualization, Data curation, Formal analysis, Investigation, Visualization, Writing – original draft, Writing – review and editing; Jane Yang, Data curation, Formal analysis, Investigation, Visualization, Writing – review and editing; Stéphanie Ratté, Conceptualization, Data curation, Formal analysis, Investigation, Visualization, Writing – original draft, Project administration, Writing – review and editing; Steven A Prescott, Conceptualization, Data curation, Supervision, Funding acquisition, Visualization, Writing – original draft, Project administration, Writing – review and editing

### Author ORCIDs

Yu-Feng Xie ⓘ http://orcid.org/0000-0001-8384-4177
Jane Yang ⓘ https://orcid.org/0000-0003-0114-5503
Stéphanie Ratté ⓘ http://orcid.org/0000-0002-7005-6726
Steven A Prescott ⓘ https://orcid.org/0000-0002-3827-4512

### Ethics

All animal procedures were approved by the Animal Care Committee at The Hospital for Sick Children (protocol #53451) and were conducted in accordance with guidelines from the Canadian Council on Animal Care.

Reviewer #1 (Public Review): https://doi.org/10.7554/eLife.90960.3.sa1
Reviewer #2 (Public Review): https://doi.org/10.7554/eLife.90960.3.sa2
Reviewer #3 (Public Review): https://doi.org/10.7554/eLife.90960.3.sa3
Author response https://doi.org/10.7554/eLife.90960.3.sa4

## Additional files

### Supplementary files

• Supplementary file 1. Supplementary tables. A, Model data before and after channel "inhibition". B, Reagents. C, EC50 values. D, Primers. E, Model equations. F, Conductance densities at baseline for DIV 0 and DIV4-7 models.

• MDAR checklist

• Source code 1. Contains computer code for computational model of neuron.

## Data availability

All data generated or analyzed during this study are included in the manuscript and supporting files; source data files have been provided for Figures 1-3, 5-8.

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
