## [Editor Report · eLife assessment]

This **fundamental** study provides an unprecedented understanding of the roles of different combinations of NaV channel isoforms in nociceptors' excitability, with relevance for the design of better strategies targeting NaV channels to treat pain. Although the experimental combination of electrophysiological, modeling, imaging, molecular biology, and behavioral data is **convincing** and supports the major claims of the work, some results remain inconclusive and need to be strengthened by further evidence. The work may be of broad interest to scientists working on pain, drug development, neuronal excitability, and ion channels.

---

## [Referee Report · Reviewer #1 (Public Review)]

Summary:

In this work, Xie, Prescott and colleagues have reevaluated the role of Nav1.7 in nociceptive sensory neurons excitability. They find that nociceptors can make use of different sodium channel subtypes to reach equivalent excitability. The existence of this degeneracy is critical to understanding the neuronal physiology under normal and pathological conditions and could explain why Nav subtype-selective drugs have failed in clinical trials. More concretely, nociceptor repetitive spiking relies on Nav1.8 at DIV0 (and probably under normal conditions in vivo), but on Nav1.7 and Nav1.3 at DIV4-7 (and after inflammation in vivo).

The conclusions of this paper are mostly well supported by data, and these findings should be of broad interest to scientists working on pain, drug development, neuronal excitability and ion channels.

Strengths:

The authors have employed elegant electrophysiology experiments (including specific pharmacology and dynamic clamp) and computational simulations to study the excitability of a subpopulation of DRGs that would very likely match with nociceptors (they take advantage of using transgenic mice to detect Nav1.8-expressing neurons). They make a strong point showing the degeneracy that occurs at the ion channel expression level in nociceptors, adding this new data to previous observations in other neuronal types. They also demonstrate that the different Nav subtypes functionally overlap and are able to interchange their "typical" roles in action potential generation. As Xie, Prescott and colleagues argue, the functional implications of the degenerate character of nociceptive sensory neurons excitability need to be seriously taken into account regarding drug development and clinical trials with Nav subtype-selective inhibitors.

In this revised version, the quality of the manuscript has been visibly improved. In my opinion, the questions and concerns raised by reviewers have been addressed clearly. After a detailed reading of this version and the comments to the reviewers, I have no additional comments or criticisms.

---

## [Referee Report · Reviewer #2 (Public Review)]

Summary:

The authors have noted in preliminary work that tetrodotoxin (TTX), which inhibits NaV1.7 and several other TTX-sensitive sodium channels, has differential effects on nociceptors, dramatically reducing their excitability under certain conditions but not under others. Partly because of this coincidental observation, the aim of the present work was to re-examine or characterize the role of NaV1.7 in nociceptor excitability and the effects on drug efficacy. The manuscript demonstrates that a NaV1.7-selective inhibitor produces analgesia only when nociceptor excitability is based on NaV1.7. More generally and comprehensively, the results show that nociceptors can achieve equivalent excitability through changes in differential NaV inactivation and NaV expression of different NaV subtypes (NaV 1.3/1.7 and 1.8). This can cause widespread changes in the role of a particular subtype over time. The degenerate nature of nociceptor excitability shows functional implications that make the assignment of pathological changes to a particular NaV subtype difficult or even impossible.

Thus, the analgesic efficacy of NaV1.7- or NaV1.8-selective agents depends essentially on which NaV subtype controls excitability at a given time point. These results explain, at least in part, the poor clinical outcomes with the use of subtype-selective NaV inhibitors and therefore have major implications for the future development of Nav-selective analgesics.

Strengths:

The results are clearly and impressively supported by the experiments and data shown. During the revision, the manuscript was consistently improved and the concerns of the first reviews were resolved. All methods are described in detail, and presumably, allow good reproducibility and were suitable to address the scientific question.

The results showing that nociceptors can achieve equivalent excitability through changes in differential NaV inactivation and expression of different NaV subtypes are of great importance in the fields of basic and clinical pain research and sodium channel physiology and pharmacology, but also for a broad readership and community. The degenerate nature of nociceptor excitability, which is clearly shown and well supported by data has large functional implications. The results are of great importance because they may explain, at least in part, the poor clinical outcomes with the use of subtype-selective NaV inhibitors and therefore have major implications for the future development of Nav-selective analgesics.

In summary, the authors achieved their overall aim to enlighten the role of the NaV1.7 in nociceptor excitability and the effects on drug efficacy. The data support the conclusions and clinical implications are highlighted as far as is currently justifiable due to the still limited experience in translation. This appears well-considered, not too speculative, and ultimately appropriate.

The main weaknesses of the first version were fixed during the revision:

(i) After revising the manuscript, the initial weakness that the computer model was described superficially has been fixed. Important information was added to the main text and additional information, including the full code and equations and values are deposited on ModelDB or are given in the Supplementary information (Suppl. Table 5 & 6).

(ii) The authors now comment that corresponding studies on protein levels or e.g. neuroinflammatory changes could support the characterization of the time course of membrane expression and cellular changes, but this should be addressed in future studies, as these analyses would also raise new questions, such as about membrane trafficking, post-translational modifications, etc. This is plausible and has now been appropriately addressed in the text.

(iii) During the initial review the authors were asked to discuss the promising role of NaV1.7 in the light of clinical results. In their response, the authors confidently state that they „wish to avoid speculating on which particular clinical results are better explained because our study was not designed for that." They, however, emphasize their take-home message, which is well supported "Instead, our take-home message (which is well supported; see Discussion on lines 309-321) is that NaV1.7-selective drugs may have a variable clinical effect because nociceptors' reliance on NaV1.7 is itself variable - much more than past studies would have readers believe. ... The challenge (as highlighted in the Abstract, lines 21-22) is that identifying the dominant Nav subtype to predict drug efficacy is difficult."

Against the background of this argumentation, it must be admitted that the decision not to present as yet unproven speculations is probably appropriate from a scientific point of view and that this ultimately proves the critical assessment of one's own data and the limitations of the study. This is undoubtedly acceptable and - in retrospect - probably the right way to go.

---

## [Referee Report · Reviewer #3 (Public Review)]

Summary:

In this study the authors used patch-clamp to characterize the implication of various voltage-gated Na+ channels in the firing properties of mouse nociceptive sensory neurons. They claim that depending on the culture conditions NaV1.3, NaV1.7, and NaV1.8 have distinct contributions to action potential firing and that similar firing patterns can result from distinct relative roles of these channels.

Strengths:

The paper addresses the important issue of understanding the lack of success of therapeutic strategies targeting NaV channels in the context of pain. Specifically, the authors test the hypothesis that different NaV channels contribute in a plastic manner to action potential firing, which may be the reason why it is difficult to target pain by inhibiting these channels.

Weaknesses:

(1) - The main claim of this paper is that "nociceptors can achieve equivalent excitability using different combinations of NaV1.3, NaV1.7, and NaV1.8". From this, they allude to the manifestation of "degeneracy", a concept implying that a biological process can occur via distinct sets of underlying components.

In my opinion, the analyses of the data is biased towards the author's interpretation.

- First, when comparing the excitability across neurons one should relate the response (in this case mean firing frequency) to the absolute size of the stimulus, not to the size of the stimulus normalized to the rheobase (see e.g., Figs. 1A). From this particular figure the authors conclude that the excitability is similar in the culture stages DIV0 and DIV4-7, but these data were not directly compared.

- Second, the authors reach their conclusion from the comparison of the (average) firing rate determined over 1 s current stimulation in distinct conditions. However, this is not the only parameter that determines how sensory neurons might convey information. For instance, the time dependence of the instantaneous frequency, the actual firing pattern, maybe also important.

- Third, the use of 1 s of current stimulation might not be sufficient to characterize the firing pattern if one wants to obtain conclusions that could translate to clinical settings (i.e., sustained pain).

- Fourth, out of principle, the gating properties of NaV1.7 and NaV1.8 channels are not identical, and therefore their contributions to excitability should not be the same. A neuron in which NaV1.7 is the main contributor is expected to have a damping firing pattern due to cumulative channel inactivation, whereas another depending mainly on NaV1.8 is expected to display more sustained firing. This is actually seen in the results of the modelling.

(2) - The quality of some recordings is dubious. The currents shown as TTX-sensitive in Fig. 1D look very strange (not like the ones at Baseline DIV4-7). These traces show abnormally fast inactivation and even transient deflections above zero current line. These are obvious artifacts of the subtraction procedure, probably due to unstable current amplitudes along the recording time. Similar odd-looking traces are shown in Fig. 3A.

(3) - I would like to point out that the main Significance Statement of the manuscript reads "The analgesic efficacy of subtype-selective drugs hinges on which subtype controls excitability". I would like to point out that, in addition of being extremely obvious for anyone knowing a bit about pain signaling, the authors did not test the analgesic efficacy of any drug in this study.

(4) - A critical issue in the manuscript is the unnecessary use of phrases that imply that biological entities have some sort of willpower, flirting with anthropomorphism and teleological language.

Sentences such as "Nociceptive sensory neurons convey pain signals to the CNS using action potentials" (see the Abstract) should be avoided. Neurons do not really "use" action potentials, they have no will to do so. Action potentials are not tools or means to be "used" by neurons. There are many other examples of misuse of the verb "use" in many other sentences. These were pointed out during the revision phase, but unfortunately the authors refused to correct them.

---

## [Author Response]

The following is the authors’ response to the original reviews.

**eLife assessment**
This fundamental study provides an unprecedented understanding of the roles of different combinations of NaV channel isoforms in nociceptors' excitability, with relevance for the design of better strategies targeting NaV channels to treat pain. Although the experimental combination of electrophysiological, modeling, imaging, molecular biology, and behavioral data is convincing and supports the major claims of the work, some conclusions need to be strengthened by further evidence or discussion. The work may be of broad interest to scientists working on pain, drug development, neuronal excitability, and ion channels.
**Reviewer #1 (Public Review):**
Summary:In this work, Xie, Prescott, and colleagues have reevaluated the role of Nav1.7 in nociceptive sensory neuron excitability. They find that nociceptors can make use of different sodium channel subtypes to reach equivalent excitability. The existence of this degeneracy is critical to understanding neuronal physiology under normal and pathological conditions and could explain why Nav subtype-selective drugs have failed in clinical trials. More concretely, nociceptor repetitive spiking relies on Nav1.8 at DIV0 (and probably under normal conditions in vivo), but on Nav1.7 and Nav1.3 at DIV4-7 (and after inflammation in vivo).The conclusions of this paper are mostly well supported by data, and these findings should be of broad interest to scientists working on pain, drug development, neuronal excitability, and ion channels.Strengths:(1.1) The authors have employed elegant electrophysiology experiments (including specific pharmacology and dynamic clamp) and computational simulations to study the excitability of a subpopulation of DRGs that would very likely match with nociceptors (they take advantage of using transgenic mice to detect Nav1.8-expressing neurons). They make a strong point showing the degeneracy that occurs at the ion channel expression level in nociceptors, adding this new data to previous observations in other neuronal types. They also demonstrate that the different Nav subtypes functionally overlap and are able to interchange their "typical" roles in action potential generation. As Xie, Prescott, and colleagues argue, the functional implications of the degenerate character of nociceptive sensory neuron excitability need to be seriously taken into account regarding drug development and clinical trials with Nav subtype-selective inhibitors.Weaknesses:(1.2) The next comments are minor criticisms, as the major conclusions of the paper are well substantiated. Most of the results presented in the article have been obtained from experiments with DRG neuron cultures, and surely there is a greater degree of complexity and heterogeneity about the degeneracy of nociceptors excitability in the "in vivo" condition. Indeed, the authors show in Figures 7 and 8 data that support their hypothesis and an increased Nav1.7's influence on nociceptor excitability after inflammation, but also a higher variability in the nociceptors spiking responses. On the other hand, DRG neurons targeted in this study (YFP (+) after crossing with Nav1.8-Cre mice) are >90% nociceptors, but not all nociceptors express Nav1.8 in vivo. As shown by Li et al., 2016 ("Somatosensory neuron types identified by high-coverage single-cell RNA-sequencing and functional heterogeneity"), there is a high heterogeneity of neuron subtypes within sensory neurons. Therefore, some caution should be taken when translating the results obtained with the DRG neuron cultures to the more complex "in vivo" panorama.

We agree that most but not all Nav1.8+ DRG cells are nociceptors and that not all nociceptors express Nav1.8. We targeted small neurons that also express (or at some point expressed) Nav1.8, thus excluding larger neurons that express Nav1.8. This allowed us to hone in on a relatively homogeneous set of neurons, which is crucial when testing different neurons to compare between conditions (as opposed to testing longitudinally in the same neuron, which is not feasible). We expect all neurons are degenerate but likely on the basis of different ion channel combinations. Indeed, even within small Nav1.8+ neurons, other channels that we did not consider likely contribute to the degenerate regulation (as now better reflected in the revised Discussion).

That said, there are multiple sources of heterogeneity. We suspect that heterogeneity is more increased after inflammation than after axotomy because all DRG neurons experience axotomy when cultured whereas neurons experience inflammation differently in vivo depending on whether their axon innervates the inflamed area (now explained on lines 214-215). This is not so much about whether the insult occurs in vivo or in vitro, but about how homogeneously neurons are affected by the insult. Granted, neurons are indeed more likely to be heterogeneously affected in vivo since conditions are more complex. But our goal in testing PF-71 in behavioral tests (Fig. 8) was to show that changes observed in nociceptor excitability in Figure 7, despite heterogeneity, were predictive of changes in drug efficacy. In short, we establish Nav interchangeability by comparing neurons in culture (Figs 1-6), but we then show that similar Nav shifts can develop in vivo (Fig 7) with implications for drug efficacy (Fig 8). Such results should alert readers to the importance of degeneracy for drug efficacy (which is our main goal) even without a complete picture of nociceptor degeneracy or DRG neuron heterogeneity. Additions to the Discussion (lines 248-259, 304-308) are intended to highlight these considerations.

(1.3) Although the authors have focused their attention on Nav channels, it should be noted that degeneracy concerning other ion channels (such as potassium ion channels) could also impact the nociceptor excitability. The action potential AHP in Figure 1, panel A is very different comparing the DIV0 (blue) and DIV4-7 examples. Indeed, the conductance density values for the AHP current are higher at DIV0 than at DIV7 in the computational model (supplementary table 5). The role of other ion channels in order to obtain equivalent excitability should not be underestimated.

We completely agree. We focused on Nav channels because of our initial observation with TTX and because of industry’s efforts to develop Nav subtype-selective inhibitors, whose likelihood of success is affected by the changes we report. But other channels are presumably changing, especially given observed changes in the AHP shape (now mentioned on lines 304-308).Investigation should be expanded to include these other channels in future studies.

**Reviewer #2 (Public Review):**
Summary:The authors have noted in preliminary work that tetrodotoxin (TTX), which inhibits NaV1.7 and several other TTX-sensitive sodium channels, has differential effects on nociceptors, dramatically reducing their excitability under certain conditions but not under others. Partly because of this coincidental observation, the aim of the present work was to re-examine or characterize the role of NaV1.7 in nociceptor excitability and its effects on drug efficacy. The manuscript demonstrates that a NaV1.7-selective inhibitor produces analgesia only when nociceptor excitability is based on NaV1.7. More generally and comprehensively, the results show that nociceptors can achieve equivalent excitability through changes in differential NaV inactivation and NaV expression of different NaV subtypes (NaV 1.3/1.7 and 1.8). This can cause widespread changes in the role of a particular subtype over time. The degenerate nature of nociceptor excitability shows functional implications that make the assignment of pathological changes to a particular NaV subtype difficult or even impossible.Thus, the analgesic efficacy of NaV1.7- or NaV1.8-selective agents depends essentially on which NaV subtype controls excitability at a given time point. These results explain, at least in part, the poor clinical outcomes with the use of subtype-selective NaV inhibitors and therefore have major implications for the future development of Nav-selective analgesics.Strengths:(2.1) The above results are clearly and impressively supported by the experiments and data shown. All methods are described in detail, presumably allow good reproducibility, and were suitable to address the corresponding question. The only exception is the description of the computer model, which should be described in more detail.

We failed to report basic information such as the software, integration method and time step in the original text. This information is now provided on lines 476-477. Notably, the full code is available on ModelDB plus all equations including the values for all gating parameters are provided in Supplementary Table 5 and values for maximal conductance densities for DIV0 and DIV7 models are provided in Supplementary Table 6. Changes in conductance densities to simulate different pharmacological conditions are reported in the relevant figure legends (now shown in red). We did not include model details in the main text to avoid disrupting the flow of the presentation, but all the model details are reported in the Methods, tables and/or figure legends.

(2.2) The results showing that nociceptors can achieve equivalent excitability through changes in differential NaV inactivation and expression of different NaV subtypes are of great importance in the fields of basic and clinical pain research and sodium channel physiology and pharmacology, but also for a broad readership and community. The degenerate nature of nociceptor excitability, which is clearly shown and well supported by data has large functional implications. The results are of great importance because they may explain, at least in part, the poor clinical outcomes with the use of subtype-selective NaV inhibitors and therefore have major implications for the future development of Nav-selective analgesics.In summary, the authors achieved their overall aim to enlighten the role of NaV1.7 in nociceptor excitability and the effects on drug efficacy. The data support the conclusions, although the clinical implications could be highlighted in a more detailed manner.Weaknesses:As mentioned before, the results that nociceptors can achieve equivalent excitability through changes in differential NaV inactivation and NaV expression of different NaV subtypes are impressive. However, there is some "gap" between the DRG culture experiments and acutely dissociated DRGs from mice after CFA injection. In the extensive experiments with cultured DRG neurons, different time points after dissociation were compared. Although it would have been difficult for functional testing to examine additional time points (besides DIV0 and DIV47), at least mRNA and protein levels should have been determined at additional time points (DIV) to examine the time course or whether gene expression (mRNA) or membrane expression (protein) changes slowly and gradually or rapidly and more abruptly.

Characterizing the time course of NaV expression changes is worthwhile but, insofar as such details are not necessary to establish that excitability is degenerate, it was not include in the current study. Furthermore, since mRNA levels do not parallel the functional changes in Nav1.7 (Figure 6A), we do not think it would be helpful to measure mRNA levels at intermediate time points. Measuring protein levels would be more informative, however, as now explained on lines 362-369, neurons were recorded at intermediate time points in initial experiments and showed a lot of variability. Methods that could track fluorescently-tagged NaV channels longitudinally (i.e. at different time points in the same cell) would be well suited for this sort of characterization, but will invariably lead to more questions about membrane trafficking, phosphorylation, etc. We agree that a thorough characterization would be interesting but we think it is best left for a future study.

It would also be interesting to clarify whether the changes that occur in culture (DIV0 vs. DIV47) are accompanied by (pro-)inflammatory changes in gene and protein expression, such as those known for nociceptors after CFA injection. This would better link the following data demonstrating that in acutely dissociated nociceptors after CFA injection, the inflammationinduced increase in NaV1.7 membrane expression enhances the effect of (or more neurons respond to) the NaV1.7 inhibitor PF-71, whereas fewer CFA neurons respond to the NaV1.8 inhibitor PF-24.

These are some of the many good questions that emerge from our results. We are not particularly keen to investigate what happens over several days in culture, since this is not so clinically relevant, but it would be interesting to compare changes induced by nerve injury in vivo (which usually involves neuroinflammatory changes) and changes induced by inflammation. Many previous studies have touched on such issues but we are cautious about interpreting transcriptional changes, and of course all of these changes need to be considered in the context of cellular heterogeneity. It would be interesting to decipher if changes in NaV1.7 and NaV1.8 are directly linked so that an increase in one triggers a decrease in the other, and vice versa. But of course many other channels are also likely to change (as discussed above), and they too warrant attention, which makes the problem quite difficult. We look forward to tackling this in future work.

The results shown explain, at least in part, the poor clinical outcomes with the use of subtypeselective NaV inhibitors and therefore have important implications for the future development of Nav-selective analgesics. However, this point, which is also evident from the title of the manuscript, is discussed only superficially with respect to clinical outcomes. In particular, the promising role of NaV1.7, which plays a role in nociceptor hyperexcitability but not in "normal" neurons, should be discussed in light of clinical results and not just covered with a citation of a review. Which clinical results of NaV1.7-selective drugs can now be better explained and how?

We wish to avoid speculating on which particular clinical results are better explained because our study was not designed for that. Instead, our take-home message (which is well supported; see Discussion on lines 309-321) is that NaV1.7-selective drugs may have a variable clinical effect because nociceptors’ reliance on NaV1.7 is itself variable – much more than past studies would have readers believe. At the end of the results (line 235), which is, we think, what prompted the reviewer’s comment, we point to the Discussion. The corollary is that accounting for degeneracy could help account for variability in drug efficacy, which would of course be beneficial. The challenge (as highlighted in the Abstract, lines 21-22) is that identifying the dominant Nav subtype to predict drug efficacy is difficult. We certainly don’t have all the answers, but we hope our results will point readers in a new direction to help answer such questions.

Another point directly related to the previous one, which should at least be discussed, is that all the data are from rodents, or in this case from mice, and this should explain the clinical data in humans. Even if "impediment to translation" is briefly mentioned in a slightly different context, one could (as mentioned above) discuss in more detail which human clinical data support the existence of "equivalent excitability through different sodium channels" also in humans.

We are not aware of human data that speak directly to nociceptor degeneracy but degeneracy has been observed in diverse species; if anything, human neurons are probably even more degenerate based on progressive expansion of ion channel types, splice variants, etc. over evolution. Of course species differences extend beyond degeneracy and are always a concern for translation, because of a species difference in the drug target itself or because preclinical pain testing fails to capture the most clinically important aspects of pain (which we mention on line 35). Line 39 now reiterates that these explanations for translational difficulties are not mutually exclusive, but that degeneracy deserves greater consideration that is has hitherto received. Indeed, throughout our paper we imply that degeneracy may contribute to the clinical failure of Nav subtype-specific drugs, but those failures are certainly not evidence of degeneracy. In the Discussion (line 320-321), we now cite a recent review article on degeneracy in the context of epilepsy, and point out how parallels might help inform pain research. We wish we had a more direct answer to the reviewer’s request; in the absence of this, we hope our results motivate readers to seek out these answers in future research.

Although speculative, it would be interesting for readers to know whether a treatment regimen based on "time since injury" with NaV1.7 and NaV1.8 inhibitors might offer benefits. Based on the data, could one hypothesize that NaV1.7 inhibitors are more likely to benefit (albeit in the short term) in patients with neuropathic pain with better patient selection (e.g., defined interval between injury and treatment)?

We like that our data prompt this sort of prediction. However, this is potentially complicated since the injury may be subtle, which is to say that the exact timing may not be known. There are scenarios (e.g. postoperative pain) where the timing of the insult is known, but in other cases (e.g. diabetic neuropathy) the disease process is quite insidious, and different neurons might have progressed through different stages depending on how they were exposed to the insult. Our own experiments with CFA are a case in point. Notwithstanding the potential difficulties about gauging the time course, any way of predicting which Nav subtype is dominant could help more strategically choose which drug to use.

**Reviewer #3 (Public Review):**
Summary:In this study, the authors used patch-clamp to characterize the implication of various voltagegated Na+ channels in the firing properties of mouse nociceptive sensory neurons. They report that depending on the culture conditions NaV1.3, NaV1.7, and NaV1.8 have distinct contributions to action potential firing and that similar firing patterns can result from distinct relative roles of these channels. The findings may be relevant for the design of better strategies targeting NaV channels to treat pain.Strengths:The paper addresses the important issue of understanding, from an interesting perspective, the lack of success of therapeutic strategies targeting NaV channels in the context of pain. Specifically, the authors test the hypothesis that different NaV channels contribute in a plastic manner to action potential firing, which may be the reason why it is difficult to target pain by inhibiting these channels. The experiments seem to have been properly performed and most conclusions are justified. The paper is concisely written and easy to follow.Weaknesses:(1) The most critical issue I find in the manuscript is the claim that different combinations of NaV channels result in equivalent excitability. For example, in the Abstract it is stated that: "...we show that nociceptors can achieve equivalent excitability using different combinations of NaV1.3, NaV1.7, and NaV1.8". The gating properties of these channels are not identical, and therefore their contributions to excitability should not be the same. I think that the culprit of this issue is that the authors reach their conclusion from the comparison of the (average) firing rate determined over 1 s current stimulation in distinct conditions. However, this is not the only parameter that determines how sensory neurons convey information. For instance, the time dependence of the instantaneous frequency, the actual firing pattern, may be important too. Moreover, the use of 1 s of current stimulation might not be sufficient to characterize the firing pattern if one wants to obtain conclusions that could translate to clinical settings (i.e., sustained pain). A neuron in which NaV1.7 is the main contributor is expected to have a damping firing pattern due to cumulative channel inactivation, whereas another depending mainly on NaV1.8 is expected to display more sustained firing. This is actually seen in the results of the modelling.

This concern seems to boil down to how equivalent is equivalent? The spike shape or the full inputoutput curve for a DIV0 neuron (Nav1.8-dominant) is never equivalent to what’s seen in a DIV47 neuron (Nav1.7-dominant), but nor are any two DIV0 neurons strictly equivalent, and likewise for any two DIV4-7 neurons. Our point is that DIV0 and DIV4-7 neurons are a far more similar (less discriminable) in their excitability than expected from the qualitative difference in their TTX sensitivity (and from repeated claims in the literature that Nav1.7 is necessary for spike generation in nociceptors). Nav isoforms need not be identical to operate similarly; for instance, Nav1.8 tends to activate at “suprathreshold” voltages, but this depends on the value of threshold; if threshold increases, Nav1.8 can activate at subthreshold voltages (see Fig 5). We have modified lines 155- 175 to help clarify this.

We completely agree that firing rate is not the only way to convey sensory information, and of course injecting current directly into the cell body via a patch pipette is not a natural stimulus. These are all factors to keep in mind when interpreting our data. Nonetheless, our data show that excitability is similar between DIV0 and DIV 4-7, so much so that data from any one neuron (without pharmacological tests or capacitance measurements) would likely not reveal if that cell is DIV0 or DIV4-7; this “indiscriminability” qualifies as “equivalent” for our purposes, and is consistent with phrasing used by other authors studying degeneracy. Notably, not every DIV4-7 neuron exhibits spike height attenuation (see Fig. 1A), likely because of concomitant changes in the AHP that were not captured in our computer model or directly tested in our experiments. This highlights that other channel changes may also contribute to degeneracy and the maintenance of repetitive spiking.

(2) In Fig. 1, is 100 nM TTX sufficient to inhibit all TTX-sensitive NaV currents? More common in literature values to fully inhibit these currents are between 300 to 500 nM. The currents shown as TTX-sensitive in Fig. 1D look very strange (not like the ones at Baseline DIV4-7). It seems that 100 nM TTX was not enough, leading to an underestimation of the amplitude of the TTXsensitive currents.

As now summarized in Supplementary Table 3 (which is newly added), 100 nM TTX is >20x the EC50 for Nav1.3 and Nav1.7 (but is still far below the EC50 for Nav1.8). Based on this, TTXsensitive channels are definitely blocked in our TTX experiments.

(3) Page 8, the authors conclude that "Inflammation caused nociceptors to become much more variable in their reliance of specific NaV subtypes". However, how did the authors ensure that all neurons tested were affected by the CFA model? It could be that the heterogeneity in neuron properties results from distinct levels of effects of CFA.

We agree with the reviewer. We also believe that variable exposure to CFA is the most likely explanation for the heightened variability in TTX-sensitivity reported in Figure 7 (now more clearly explained on lines 214-215). One could try co-injecting a retrograde dye with the CFA to label cells innervating the injection site, but differential spread of the CFA and dye are liable to preclude any good concordance. Alternatively, a pain model involving more widespread (systemic) inflammation might cause a more homogeneous effect. But, our main goal with CFA injections was to show that a Nav1.8Nav1.7 switch can occur in vivo (and is therefore not unique to culturing), and that demonstration is true even if some neurons do not switch. Subsequent testing in Figure 8 shows that enough neurons switch to have a meaningful effect in terms of the behavioral pharmacology. So, notwithstanding tangential concerns, we think our CFA experiments succeeded in showing that Nav channels can switch in vivo and that this impacts drug efficacy.

**Recommendations for the authors:**
All reviewers agreed that these results are solid and interesting. However, the reviewers also raised several concerns that should be addressed by the authors to improve the strength of the evidence presented. Revisions considered to be essential include:(1) Discuss how degeneracy concerning other ion channels (such as potassium ion channels) could also impact nociceptor excitability (reviewer #1). Additionally, the translation of results from DRG neuron cultures to "in vivo" nociceptors should be better discussed.

We have added a new paragraph to the Discussion (line 248-259) to remind readers that despite our focus on Nav channels, other ion channels likely also change (and that these changes involve diverse regulatory mechanisms that require further investigation). Likewise, despite our focus on the changes caused by culturing neurons, we remind readers that subtler, more clinically relevant in vivo perturbations can likewise cause a multitude of changes. We end that paragraph by emphasizing that although accounting for all the contributing components is required to fully understand a degenerate system, meaningful progress can be made by studying a subset of the components. We want to emphasize this because there is some middle ground between focusing on one component at a time (which is the norm) vs. trying to account for everything (which is an infeasible ideal). Additional text on lines 304-308 also addresses related points.

(2) Discuss how different combinations of NaV channels result in equivalent excitability, in the context of the experimental conditions used (see main comment by reviewer #3). It should also be discussed in more detail which human clinical data support the existence of "equivalent excitability through different sodium channels" also in humans (reviewer #2).

Regarding the first part of this comment, reviewer 3 wrote in the public review that “The gating properties of these channels are not identical, and therefore their contributions to excitability should not be the same.” Differences in gating properties are commonly used to argue that different Nav subtypes mediate different phases of the spike, for example, that Nav1.7 initiates the spike whereas Nav1.8 mediates subsequent depolarization because Nav1.7 and Nav1.8 activate at perithreshold and suprathrehold voltages, respectively (see lines 134-135, now shown in red). But such comparison is overly simplistic insofar as it neglects the context in which ion channels operate. For instance, if Nav1.7 is not expressed or fully inactivates, voltage threshold will be less negative, enabling Nav1.8 to contribute to spike initiation; in other words, previously “suprathreshold” voltages become “perithreshold”. Figure 5 is dedicated to explaining this context-sensitivity; specifically, we demonstrate with simulations how Nav1.8 takes over responsibility for initiating a spike when Na1.7 is absent or inactivated. Text on lines 155- 184 has been edited to help clarify this. Regarding the second part of this comment, we are not aware of any direct evidence from human sensory neurons that different sodium channels produce equivalent excitability, but that is certainly what we expect. We suggest that failure of Nav subtype-specific drugs is, at least in part, because of degeneracy, but such failures do not demonstrate degeneracy unless other contributing factors can be excluded (which they can’t). Recognizing degeneracy is difficult, and so variability that might be explained by degeneracy will go unexplained or attributed to other factors unless, by design or serendipity, experiments quantify the effects of degeneracy (as we have attempted to do here). We now cite a recent review article on degeneracy and epilepsy (line 320), which addresses relevant themes that might help inform pain research; for instance, most existing antiseizure medications act on multiple targets whereas more recently developed single-target drugs have proven largely ineffective. This is similar to but better documented than for analgesics. With this in mind, we revised the text to emphasize the circumstantial nature of existing evidence and the need to test more directly for degeneracy (lines 320-323).

(3) Extend the discussion about the poor clinical outcomes with the use of subtype-selective NaV inhibitors. In particular, the promising role of NaV1.7, which plays a role in nociceptor hyperexcitability but not in "normal" neurons, should be discussed in light of clinical results and not just covered with a citation of a review. Which clinical results of NaV1.7-selective drugs can now be better explained and how? (reviewer #2)

As discussed above, we are cautious avoid speculating on which clinical results are attributable to degeneracy. Instead, our take-home message (see Discussion, lines 309-323) is that NaV1.7selective drugs may have a variable clinical effect because nociceptors’ reliance on NaV1.7 is itself variable – much more than past studies would have readers believe. The corollary is that accounting for degeneracy could help account for variability in drug efficacy, which would of course be beneficial. The challenge (as highlighted in the Abstract, lines 21-22) is that identifying the dominant Nav subtype to predict drug efficacy is not trivial. Interpreting clinical data is also complicated by the fact that we are either dealing with genetic mutations (with unclear compensatory changes) or pharmacological results (where NaV1.7-selective drugs have a multitude of problems that might contribute to their lack of efficacy, separate from effects of degeneracy). We have striven to contextualize our results (e.g. last paragraph of results, lines 222-235). We think this is the most we can reasonably say based on the limitations of existing clinical data.

(4) Provide a clearer and more detailed description of the computational model (reviewers #2 and #3).

We added important details on line 476-477 but, in our honest opinion, we think our computational model is thoroughly explained. The issue seems to boil down to whether details are included in the Results vs. being left for the Methods, tables and figure legends. We prefer the latter.

(5) Better clarify the effects of the CFA model, to provide further evidence relating inflammation with nociceptors variability (reviewers #2 and #3)

As explained in response to a specific point by reviewer #3, we believe that variable exposure to CFA explains the heightened variability in TTX-sensitivity reported in Figure 7 (now explained on lines 214-215). One could try co-injecting a retrograde dye with the CFA to label cells innervating the injection site, but differential spread of the inflammation and dye are liable to preclude any good concordance. Alternatively, a pain model involving more widespread (systemic) inflammation might cause a more homogeneous effect. But, our main goal with CFA injections was to show that a Nav1.8Nav1.7 switch can occur in vivo (and is therefore not unique to culturing); that demonstration holds true even if some neurons do not switch. Subsequent testing (Fig 8) shows that enough neurons switch to drug efficacy assessed behaviorally. This is emphasized with new text on lines 225-227. Overall, we think our CFA experiments succeed in showing that Nav channels can switch in vivo and, despite variability, that this occurs in enough neurons to impact drug efficacy.

(6) Revise the text according to all recommendations raised by the reviewers and listed in the individual reviews.

Detailed responses are provided below for all feedback and changes to the text were made whenever necessary, as identified in our responses.

**Reviewer #1 (Recommendations For The Authors):**
Minor points/recommendations:Protein synthesis inhibition by cercosporamide could be the direct cause of a smaller-thanexpected increase in Nav1.7 levels at DIV5. But for Nav1.8, there is a mitigation in the increased levels at DIV5, that only could be explained by several indirect mechanisms, including membrane trafficking and posttranslational modifications (phosphorylation, SUMOylation, etc.) on Nav1.8 or protein regulators of Nav1.8 channels. The authors suggest that "translational regulation is crucial", but also insinuate that other processes (membrane trafficking, etc.) could contribute to the observed outcome. It is difficult to assess the relative importance of these different explanations without knowing the exact mechanisms that are acting here.

We agree. We relied on electrophysiology (and pharmacology) to measure functional changes, but we wanted to verify those data with another method. We expected mRNA levels to parallel the functional changes but, when that did not pan out, we proceeded to look at protein levels. Perhaps we should have stopped there, but by blocking protein translation, we show that there is not enough Nav1.7 protein already available that can be trafficked to the membrane. That does not explain why Nav1.8 levels drop. Our immunohistochemistry could not tease apart membrane expression from overall expression, which limits interpretation. We have enhanced the text to discuss this (lines 200-204), but further experiments are needed. Though admittedly incomplete, our initial finding help set the stage for future experiments on this matter.

Page 15, typo: "contamination from genomic RNA" -> "contamination from genomic DNA" (appears twice).

This has been corrected on lines 420 and 421.

Page 17: I could not find the computer code at ModelDB (http://modeldb.yale.edu/267560). It seems to be an old web link. It should be available at some web repository.

We confirmed that the link works. Entry is password-protected (password = excitability; see line 476). Password protection will be removed once the paper is officially published.

Page 19, reference 36, typo: "Inhibitio of" -> "Inhibition of".

This has been corrected (line 557).

Page 33, typo: "are significantly larger than differences at DIV1" -> "are significantly larger than differences at DIV0".

This has been corrected (line 796).

Page 35, figure 6 legend. The number of experiments (n) is not indicated for panel C data.

N = 3 is now reported (line 828).

**Reviewer #2 (Recommendations For The Authors):**
p. 3/4 and Data of Fig. 6: It should be commented on why days 1-3 were not investigated. An investigation of the time course (by higher frequency testing) would certainly have an added value because it would be possible to deduce whether the changes develop slowly and gradually, or whether the excitability induced by different NaVs changes suddenly. At least mRNA and protein levels should be determined at additional time points to examine the time course or whether gene expression (mRNA) or membrane expression (protein) changes slowly and gradually or rapidly and more abruptly. It would also be interesting to clarify whether the changes that occur in culture (DIV0 vs. DIV4-7) are accompanied by (pro-)inflammatory changes in gene and protein expression, such as those known for nociceptors after CFA injection. Or is the latter question clear in the literature?

We now explain (lines 362-369) that intermediate time points (DIV1-3) were tested in initial current clamp recordings. Those data showed that TTX-sensitivity stabilized by DIV4 and differed from the TTX-insensitivity observed at DIV0. TTX-sensitivity was mixed at DIV1-3 and crosscell variability complicated interpretation. Subsequent experiments were prioritized to clarify why NaV1.7 is not always critical for nociceptor excitability, contrary to past studies. Our efforts to measure mRNA and protein levels were primarily to validate our electrophysiological findings; we are also interested in deciphering the underlying regulatory processes but this is an entire study on its own. Unfortunately, the existing literature does not help or point to an explanation for the Nav1.7/1.8 shift we observed.

Our evidence that mRNA levels do not parallel functional changes argues against pursuing transcriptional changes in Nav1.7, though transcriptional changes in other factors might be important. Interpretation of immuno quantification would be complicated by the high variability we observed with the physiology at intermediate time points and, furthermore, we cannot resolve surface expression from overall expression based on available antibodies. Methods conducive to longitudinal measurements would be more appropriate (as now mentioned on line 367-369). In short, a lot more work is required to understand the mechanisms involved in the switch, but we think the existing demonstration suffices to show that NaV1.7 and NaV1.8 protein levels vary, with crucial implications for which Nav subtype controls nociceptor excitability, and important implications for drug efficacy. Explaining why and how quickly those protein levels change will be no small feat is best left for a future study.

p. 4 and following: In order to enable the interpretation of the used concentration of PF-24, PF71, and ICA, the respective IC50 should be indicated.

A table (now Supplementary Table 3; line 861) has been added to report EC50 values for all drugs for blocking NaV1.7, NaV1.8 and NaV1.3. The concentrations we used are included on that table for easy comparison.

p. 5, end of the middle paragraph: Here it should be briefly explained -for less familiar readers- why NaV1.1 cannot be causative (ICA inhibits NaV1.1 and 1.3).

We now explain (lines 117-120) that NaV1.1 is expressed almost exclusively in medium-diameter (A-delta) neurons whereas NaV1.3 is known to be upregulated in small-diameter neurons, and so the effect we observe in small neurons is most likely via blockade NaV1.3.

p. 6, lines 4/5: At least once it should read computer model instead of model.

“Computer” has been added the first time we refer to DIV0 or DIV4-7 computer models (lines 138-139)

p. 6: the difference between Fig. 4B and Fig. 4 - Figure suppl. 1 should be mentioned briefly.

We now explain (lines 150-154) that Fig. 4B involves replacing a native channel with a different virtual channel (to demonstrate their interchangeability) whereas and Fig. 4 - Figure supplement 1 involves replacing a native channel with the equivalent virtual channel (as a positive control).

p. 6/7: the text and the conclusions regarding Figure 5 are difficult to follow. Somewhat more detailed explanations of why which data demonstrate or prove something would be helpful.

The text describing Figure 5 (lines 155-175) has been revised to provide more detail.

p. 7, last sentence of the first paragraph: How is this supported by the data? Or should this sentence be better moved to the discussion?

This sentence (now lines 182-184) is designed as a transition. The first half – “a subtype’s contribution shifts rapidly (because of channel inactivation)” – summarizes the immediately preceding data (Figure 5). The second half – “or slowly (because of [changes in conductance density])” – introduces the next section. The text show in square brackets has been revised. We hope this will be clearer based on revisions to the associated text.

p. 7, second paragraph, line 3: Please delete one "at both".

Corrected

p. 7, second paragraph: Please explain why different time points (DIV4-7, DIV5, or DIV7) were used or studied.

Initial electrophysiological experiments determined that TTX sensitivity stabilized by DIV 4 (see response to opening point) and we did not maintain neurons longer than 7 days, and so neurons recorded between DIV4 and 7 were pooled. If non-electrophysiological tests were conducted on a specific day within that range, we report the specific day, but any day within the DIV4-7 range is expected to give comparable results. This is now explained on lines 365-367.

p. 8: the text regarding Fig. 7 should also include the important data (e.g. percentage of neurons showing repetitive spinking) mentioned in the legend.

This text (lines 216-220) has been revised to include the proportion of neurons converted by PF71 and PF-24 and the associated statistical results.

Fig. 1: third panel (TTX-sensitive current...) of D & Fig. 2 subpanel of A (Nav1.8 current...). These panels should be explained or mentioned in the text and/or legends.

We now explain in the figure legends (lines 708-710; 714-715; 736-738) how those currents are found through subtraction.

Fig. 2 - figure supplement 2. One might consider taking Panel A to Fig. 2 so that the comparison to DIV0 is apparent without switching to Suppl. Figs.

We left this unchanged so that Figures 2 and 3 are equivalently organized, with negative control data left to the supplemental figures. Elife formatting makes it easy to reach the supplementary figure from the main figure, so we hope this won’t be an impediment to readers.

Fig. 6 C, middle graph (graph of Nav1.7): Please re-check, whether DIV5 none vs. 24 h and none vs. 120 h are really significantly different with such a low p-value.

We re-checked the statistics and the difference pointed out by the reviewer is significant at p=0.007. We mistakenly reported p<0.001 for all comparisons, and so this p value has been corrected; all the other p values are indeed <0.001. Notably, the data are summarized as median ± quartile because of their non-Gaussian distribution; this is now explained on line 827 (as a reminder to the statement on lines 461-462). Quartiles are more comparable to SD than to SEM (in that quartiles and SD represent the distribution rather than confidence in estimating the mean, like SEM), and so medians can differ very significantly even if quartiles overlap, as in this case.

**Reviewer #3 (Recommendations For The Authors):**
(1) A critical issue in the manuscript is the use of teleological language. It is likely that this is not the intention, but careful revision of the language should be done to avoid the use of expressions that confer purpose to a biological process. Please, find below a list of statements that I consider require correction.In the Abstract, the first sentence: "Nociceptive sensory neurons convey pain signals to the CNS using action potentials". Neurons do not really "use" action potentials, they have no will or purpose to do so. Action potentials are not tools or means to be "used" by neurons. Other examples of misuse of the verb "use" are found in several other sentences:"...nociceptors can achieve equivalent excitability using different combinations of NaV1.3, NaV1.7, and NaV1.8""Flexible use of different NaV subtypes - an example of degeneracy - compromises...""Nociceptors can achieve equivalent excitability using different sodium channel subtypes" "...degeneracy - the ability of a biological system to achieve equivalent function using different components...""...nociceptors can achieve equivalent excitability using different sodium channel subtypes...""Our results show that nociceptors can achieve similar excitability using different NaV channels" "...the spinal dorsal horn circuit can achieve similar output using different synaptic weight combinations...""Contrary to the view that certain ion channels are uniquely responsible for certain aspects of neuronal function, neurons use diverse ion channel combinations to achieve similar function" "In summary, our results show that nociceptors can achieve equivalent excitability using different NaV subtypes"

“Use” can mean to put into action (without necessarily implying intention). Based on definitions of the word in various dictionaries, we feel we are well within the realm of normal usage of this term. In trying to achieve a clear and succinct writing style, we have stuck with our original word choice.

At the end of page 5 and in the legend of Fig. 7, the word "encourage" is not properly used in the sentence "The ability of NaV1.3, NaV1.7 and NaV1.8 to each encourage repetitive spiking is seemingly inconsistent with the common view...". Encouraging is really an action of humans or animals on other humans or animals.

Like for “use”, we verified our usage in various dictionaries and we do not think that most readers will be confused or disturbed by our word choice. We use “encourage” to explain that increasing NaV1.3, NaV1.7 or NaV1.8 can increase the likelihood of repetitive spiking; we avoided “cause” because the probability of repetitive spiking is not raised to 100%, since other factors must always be considered.

In the Abstract and other places in the manuscript, the word "responsibility" seems to be wrongly employed. It is true that one can say, for instance, on page 4 last paragraph "we sought to identify the NaV subtype responsible for repetitive spiking at each time point". However, to confer channels with the human quality of having "responsibility" for something does not seem appropriate. See also page 8 last paragraph, the first paragraph of the Discussion, and the three paragraphs of page 11.

Again, we must respectfully disagree with the reviewer. We appreciate that this reviewer does not like our writing style but we do not believe that our style violates English norms.

(2) In the first sentence of the Abstract, nociceptive sensory neurons do not convey "pain signals". Pain is a sensation that is generated in the brain.

“Pain” is used as an adjective for “signal” and is used to help identify the type of signal.Nonetheless, since the word count allowed for it, we now refer to “pain-related signals” (line 10).

(3) I do not see the point of plotting the firing rate as a function of relative stimulus amplitude (normalized to the rheobase, e.g., Fig. 1A bottom panels, Fig. 2B, bottom-right, Fig. 2 Supp2A right, Fig. 3 B bottom panels, etc) instead of as a function of the actual stimulus amplitude. I have the impression that this maneuver hides information. This is equivalent to plotting the current amplitudes as a function of the voltage normalized by the voltage threshold for current activation, which is obviously not done.

This is how the experiments were performed, so it would be impossible to perform the statistical analysis using the absolute amplitudes post-hoc; specifically, stimulus intensities were tested at increments defined relative to rheobase rather than in absolute terms. There are pros and cons to each approach, and both approaches are commonly used. Notably, we report the value of rheobase on the figures so that readers can, with minimal arithmetic, convert to absolute stimulus intensities. No information is hidden by our approach.

(4) On page 4 it is stated that "We show later that similar changes develop in vivo following inflammation with consequences for drug efficacy assessed behaviourally (see Fig. 8), meaning the NaV channel reconfiguration described above is not a trivial epiphenomenon of culturing". However, what happens in culture may have nothing in common with what happens in vivo during inflammation. Thus, the latter data may not serve to answer whether the culture conditions induce artifacts or not. I suggest tuning down this statement by changing "meaning" to "suggesting".

On line 97, we now write “suggesting”.

(5) Page 5, first paragraph, I miss a clear description of the mathematical models. Having to skip to the Methods section to look for the details of the models as the artifices introduced to simulate different conditions is rather inconvenient.

So as not to disrupt the flow of the presentation with methodological details, we only provide a short description of the model in the Results. We have slightly expanded this to point out that the conductance-based model is also single-compartment (line 111). We provide a very thorough description of our model in the Methods, especially considering all the details provided in Supplementary Tables 1, 5 and 6. We also report conductance densities and % changes in figure legends (lines 722, 747-748; now shown in red). This is also true for Figure 3-figure supplement 2 (lines 756-759). We tried very hard to find a good balance that we hope most readers will appreciate.

(6) Page 6, second paragraph, simulations do not serve to "measure" currents.

The sentence been revised to indicate that simulations were used to “infer” currents during different phases of the spike (line 155).

(7) Page 7, regarding the tile of the subsection "Control of changes in NaV subtype expression between DIV0 and DIV4-7", the authors measured the levels of expression, but not really the mechanisms "controlling" them. I suggest writing "changes in NaV subtype expression between DIV0 and DIV4-7"

We have removed “control of” from the section title (line 185)

(8) What was the reason for adding a noise contribution in the model?

We now explain that noise was added to reintroduce the voltage noise that is otherwise missing from simulations (line 474). For instance, in the absence of noise, membrane potential can approach voltage threshold very slowly without triggering a spike, which does not happen under realistically noisy conditions. Of course membrane potential fluctuates noisily because of stochastic channel opening and a multitude of other reasons. This is not a major issue for this study, and so we think our short explanation should suffice.

(9) Please, define the concept of degeneracy upon first mention.

Degeneracy is now succinctly defined in the abstract (line 20).